# Mitigating ion flux vortex enables reversible zinc electrodeposition

Yuhang Dai [1,2,10], Wenjia Du [2,10], Haobo Dong [1,3,10], Xuan Gao[1,2] ✉, Chang Su[4], Partha P. Paul [5,6], Bratislav Lukic[5,6], Chengyi Zhang [7], Chumei Ye [8], Jinghao Li [9], Wei Zong [2], Jianwei Li[1], Yiyang Liu[1], Alexander Rack [6], Liqiang Mai [9] ✉, Paul R. Shearing[2] ✉ & Guanjie He [1] ✉

Metal anodes hold considerable promise for high-energy-density batteries but are fundamentally limited by electrochemical irreversibility caused by uneven metal deposition and dendrite formation, which compromise battery lifespan and safety. The chaotic ion flow (or ion flux vortex) near the electrode surface, driving these instabilities, has remained elusive due to limitations in conventional techniques such as scanning electron and atomic force microscopies, which are invasive and incapable of probing internal structures of deposits. Here, we employ in-situ X-ray computed tomography (CT) to non-destructively visualize Zn deposition on LAPONITE-coated Zn anodes, thereby revealing the internal structural evolution and deposition orientation. Combined with computational fluid dynamics simulations, we demonstrate that the LAPONITE coating, with its separated positive and negative charge centers, suppresses ionic vortex formation, guiding uniform, dense, and vertically aligned Zn growth along (100) plane, thereby significantly mitigating dendrite growth. This translates into a 3.17-Ah Zn-MnO$_2$ pouch cell with stable performance over 100 cycles, offering a viable path toward scalable, high-performance metal-anode batteries.

The demand for high-capacity, reliable energy storage is fueled by ambitious global climate goals[1]. For example, the United States aims for 50% of vehicle sales to be electric by 2030, while the European Union targets a 55% reduction in greenhouse gas emissions by 2030 compared to 1990 levels[2,3]. Such targets necessitate breakthroughs in battery technologies that can deliver higher energy densities, longer cycle lives, and sustainable materials[4]. Metal negative electrodes with higher capacities than those currently used are critical to advancing high-energy-density batteries for next-generation electric vehicles and grid-scale energy storage[5]. Among the candidates, zinc (Zn) negative electrodes stand out due to their natural abundance, cost-effectiveness, high theoretical capacity (820 mAh g$^{-1}$), and compatibility with aqueous electrolytes[6–11]. The use of aqueous electrolytes further offers sustainability, intrinsic safety, and high ionic conductivity, which makes aqueous Zn-metal batteries (AZMBs) well-suited for scalable applications and fast-charging scenarios[12].

However, the practical implementation of AZMBs is still hindered by significant challenges related to non-uniform Zn deposition and

[1]Christopher Ingold Laboratory, Department of Chemistry, University College London, London WC1H 0AJ, UK. [2]Department of Engineering Science, University of Oxford, Oxford OX1 3PJ, UK. [3]School of Future Technology, South China University of Technology, Guangzhou 510641, China. [4]Nuffield College, University of Oxford, Oxford OX1 1NF, UK. [5]Henry Royce Institute, Department of Materials, The University of Manchester, Manchester M13 9PL, UK. [6]ESRF - The European Synchrotron, 71 Avenue des Martyrs, Grenoble 38000, France. [7]School of Chemical Sciences, University of Auckland, Auckland 1010, New Zealand. [8]Department of Materials Science and Metallurgy, University of Cambridge, Cambridge CB3 0FS, UK. [9]State Key Laboratory of Advanced Technology for Materials Synthesis and Processing, Wuhan University of Technology, Wuhan 430070, China. [10]These authors contributed equally: Yuhang Dai, Wenjia Du, Haobo Dong. ✉e-mail: xuan.gao@eng.ox.ac.uk; mlq518@whut.edu.cn; paul.shearing@eng.ox.ac.uk; g.he@ucl.ac.uk

dendrite formation, which compromise cycle life and battery safety[13–15]. A key origin of these failures lies in chaotic ionic flows at the electrode interface[16], which give rise to localized ion flux vortices. These vortices disrupt the uniformity of Zn nucleation and induce $Zn^{2+}$ accumulation at pre-existing deposits, thereby facilitating spatially selective growth driven by local concentration gradients and electric field effects. Vortex-like ion transport has been reported in flow-driven systems, where external instabilities cause circulating flux and uneven deposition[17]. In static electrolytes, asymmetric ion migration may similarly induce localized rotational flux—hereafter referred to as an ion flux vortex—that disrupts interfacial deposition. The outcome is progressively uneven deposition and the emergence of angular, branched dendrites. This morphological instability increases the likelihood of dendrites penetrating the separator, thus raising the risk of short-circuit failure. Addressing these issues requires both rational electrode material design and a deeper understanding of mass transport phenomena at the electrode-electrolyte interface[18].

Traditional approaches have involved constructing artificial interlayers (AILs) to homogenize the flux of $Zn^{2+}$ itself through tailored functional groups or channels[19,20]. While partially effective, these methods fall short in addressing near-surface ion flux disturbances. In $ZnSO_4$ aqueous electrolyte, for instance, $Zn^{2+}$-$SO_4^{2-}$ contact ion pairs (CIPs) form readily owing to the divalent nature of $Zn^{2+}$ ions[21]. These CIPs, which are electrically neutral but asymmetrical, behave differently from free $Zn^{2+}$ under an electric field, resulting in ion flux vortices that disrupt uniform Zn deposition[22–25]. Addressing these chaotic flows requires approaches that go beyond conventional material design to actively regulate molecular-level ion transport.

Moreover, common characterization methods like scanning electron microscopy (SEM)[26] and atomic force microscopy (AFM)[27] are invasive, often altering the sample conditions and preventing accurate capture of internal, three-dimensional (3D) evolution of deposits. These limitations hinder accurate insights into underlying mechanisms, which in turn impedes the iterative development of strategies for controlling metal deposition. In contrast, in-situ X-ray computed tomography (X-ray CT) provides a non-destructive imaging tool capable of visualizing internal structural changes[28] in 3D during Zn deposition.

In this work, we introduce a nanosilicate (LAPONITE) coating on Zn electrodes (LAPO@Zn) to modulate ion transport, suppress ionic vortices, and achieve uniform deposition. Our X-ray CT analysis reveals that the LAPONITE coating, with separated positive and negative charge centers, directs Zn growth along the (100) plane, resulting in dense, vertically aligned deposition and significantly mitigating dendrite formation compared to standard Zn. Computational fluid dynamics (CFD) simulations further corroborate the elimination of ion flux vortices with the LAPONITE coating. The experimental and modeling results establish a foundation for further optimizing metal negative electrode interfaces for practical and scalable battery technologies.

## Results

### Structural feature of LAPONITE

LAPONITE, a commercially available material with distinct positive and negative charge centers, was selected to demonstrate its ability to selectively permit $Zn^{2+}$ passage and block $SO_4^{2-}$, contributing to a more uniform $Zn^{2+}$ flux. Basic structural information on LAPONITE is presented in Supplementary Fig. 1 and Supplementary Note 1. X-ray photoelectron spectroscopy (XPS) and X-ray diffraction (XRD) results confirmed its composition as $Na_{0.2}(Mg,Li)_3Si_4O_{10}(OH)_2 \cdot 4H_2O$ in a pure phase. Additionally, combined SEM (Supplementary Fig. 1), AFM, and Kelvin probe force microscope (KPFM) (Fig. 1a) investigations revealed that LAPONITE consists of stacked nanosheets with a noticeable potential increase at the edges and a decrease in the in-plane layer. Density functional theory (DFT) calculations further supported these

results, indicating electron enrichment on the LAPONITE surface (Supplementary Fig. 2), leading to a negative potential that attracts $Zn^{2+}$ and repels $SO_4^{2-}$, while the edge regions generate a positive potential that promotes $SO_4^{2-}$ adsorption and $Zn^{2+}$ repulsion. This unique structural feature allows $Zn^{2+}$ to pass through the channels formed by the stacked LAPONITE lamella while anchoring $SO_4^{2-}$ at the channel exterior, effectively separating $Zn^{2+}$ from $SO_4^{2-}$ (Fig. 1b).

After applying a LAPONITE coating onto Zn foils (LAPO@Zn), we investigated the electrodes' structural changes during cycling. Following one cycle of deposition in a symmetric cell, the LAPONITE layer was removed for characterization. The optical image in Fig. 1c shows a uniform surface on the Zn foil, without noticeable black Zn dendrites or aggregated silver-gray zinc hydroxide sulfate (ZHS), a basic zinc salt by-product. Time-of-flight secondary ion mass spectrometry (TOF-SIMS) was then used to analyze the Zn foil and the peeled-off LAPONITE. The 3D depth profiling from TOF-SIMS shows Zn mainly in the lower portion of the LAPONITE, while $SO_4^{2-}$ is in the upper part (Fig. 1c and Supplementary Fig. 3), consistent with the LAPONITE's role described in Fig. 1b. This spatial distribution is further corroborated by zeta potential measurements (Supplementary Table 1). The observed ion distribution suggests that LAPONITE has selective interaction capabilities, which results in a higher $Zn^{2+}$ transference number (the fraction of total ionic conductivity contributed by $Zn^{2+}$) for LAPO@Zn (0.82) compared to bare Zn (0.33) (Fig. 1d and Supplementary Fig. 4), indicating that the current source is primarily Zn deposition rather than $SO_4^{2-}$-related capacitance or ZHS formation. More importantly, the separation of $Zn^{2+}$ from $SO_4^{2-}$ influences the behavior of near-surface $Zn^{2+}$-$SO_4^{2-}$ CIPs. The resulting imbalance in mass transfer near the surface induces significant ion flux vortices and disrupts the uniformity of Zn deposition sites, as illustrated conceptually in the schematic streamlines of Fig. 2a. This phenomenon is attributed to the distinct response of these neutrally but asymmetrically charged CIPs to the electric field, in contrast to $Zn^{2+}$ alone (Fig. 2b). LAPONITE effectively reduces these $Zn^{2+}$-$SO_4^{2-}$ CIPs in the diffusion pathway, mitigating ion flux vortices and promoting a more uniform $Zn^{2+}$ flux and deposition (Fig. 2c). This mechanistic insight into CIP regulation and vortex suppression motivates a quantitative exploration of interfacial ion behavior via CFD simulations.

### Joint CFD-electrochemical analysis

To comparatively assess near-surface ion flux distribution, we established a dynamic simulation model through CFD method (Supplementary Fig. 5). Based on this model, we calculated the ionic concentration field on bare and coated electrode surfaces, as shown in Supplementary Fig. 6. Under pristine conditions, the LAPONITE coating, due to its $SO_4^{2-}$ adsorption capability, exhibited a superior effect in homogenizing $Zn^{2+}$ flux compared to bare Zn (Fig. 2d, e) and general coatings that lack ion sieving capabilities (Supplementary Fig. 7c). The $N/N_{avg}$ curves ($N$ and $N_{avg}$ represent the local and average $Zn^{2+}$ flux densities, respectively) in Fig. 2e show that the ratio of $Zn^{2+}$ flux at various locations on the LAPONITE surface to the average flux is closer to 1 compared to the other two samples (Fig. 2d and Supplementary Fig. 7d). Specifically, the residual sum of squares (RSS) value for LAPONITE (relative to the dotted line representing an $N/N_{avg}$ average value of 1) is 11.82, while the RSS values for bare Zn and general coatings are 40.71 and 18.31, respectively. These results indicate that changes in the $Zn^{2+}$/$SO_4^{2-}$ concentration field can significantly affect ion movement, as visualized through the streamlines. This comparison underscores the reduction of ion vortices owing to the elimination of $Zn^{2+}$-$SO_4^{2-}$ CIPs, resulting in more uniform Zn deposition sites (Fig. 2c) for LAPO@Zn case. Moreover, the ion flux vortex on bare Zn becomes more pronounced after deposition begins, whereas it remains suppressed on LAPO@Zn (Supplementary Fig. 8).

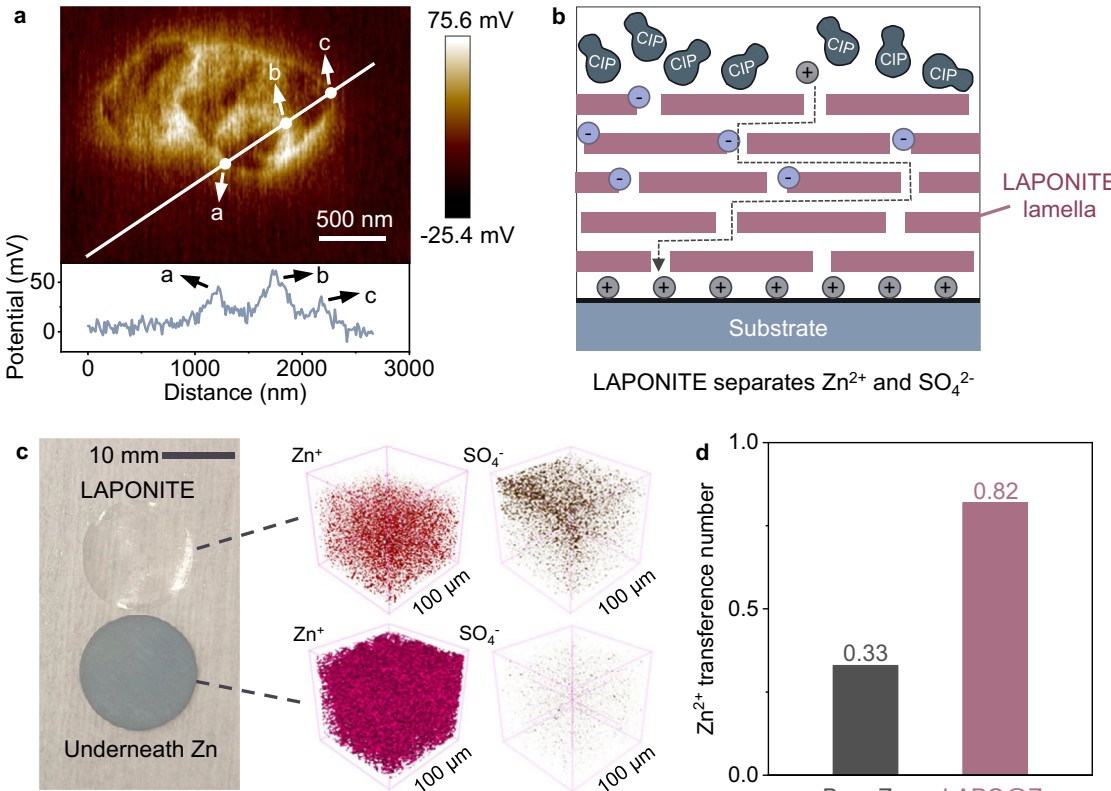

**Fig. 1 | Cation/anion separation effect of LAPONITE. a** AFM image and corresponding electrical potential curve of LAPONITE lamella. **b** Schematic illustration of the penetration of ions through the LAPONITE interlayer on substrates. The gray, periwinkle and muted navy balls represent $Zn^{2+}$, $SO_4^{2-}$, and $Zn^{2+}$-$SO_4^{2-}$ contact ion pair (CIP), respectively. **c** Optical image of the peeled LAPONITE interlayer and Zn foil underneath after cycling, together with associated 3D TOF-SIMS images. **d** The transference number of $Zn^{2+}$ on bare Zn and LAPO@Zn electrodes. Source data are provided as a Source Data file.

Beyond CFD analysis, the effect of LAPONITE on Zn deposition kinetics was thoroughly investigated. Supplementary Fig. 9 shows that polarity pairs on LAPO@Zn, arising from electron localization and surface potential variation, generate a polar electric field that directs ion movement. DFT simulations (Supplementary Fig. 10) further revealed electron localization on the LAPONITE surface, promoting selective $Zn^{2+}$ adsorption compared to the delocalized electrons on the bare Zn. These results suggest that LAPONITE facilitates electron aggregation and accelerates Zn deposition. Experimentally, electrochemical impedance spectroscopy (EIS) measurements at different temperatures demonstrated that LAPO@Zn has a much lower activation energy for interfacial charge transfer (33.32 kJ mol⁻¹) than bare Zn (46.89 kJ mol⁻¹) (Fig. 2f and Supplementary Fig. 11), implying enhanced charge transfer kinetics. Besides, galvanostatic tests demonstrated a reduced nucleation overpotential for LAPO@Zn (46.1 mV) versus bare Zn (52.6 mV) (Supplementary Fig. 12), magnifying facilitated Zn nucleation. Rate performance tests conducted on Zn ‖ Zn symmetric cells (Supplementary Fig. 13) further revealed a higher exchange current density for LAPO@Zn (16.36 mA cm⁻²) compared to bare Zn (8.95 mA cm⁻²), signifying enhanced kinetics and reduced ion flux vortices near the surface.

Chronoamperometry (CA) provides further insight into the dynamic Zn deposition process (Fig. 2g, h). During deposition, the current response can be decomposed into double layer (DL), instantaneous two-dimensional growth (2DI), and instantaneous/progressive three-dimensional growth (3DI/3DP) components in our study, described by the following equations:

$$I_{DL} = I_0 \cdot e^{-\frac{t}{\tau_{DL}}} \tag{1}$$

$$I_{2DI} = ate^{-bt^2} \tag{2}$$

$$I_{3DI} = c(1 - e^{-dt^2}) \tag{3}$$

$$I_{3DP} = c(1 - e^{-dt^3}) \tag{4}$$

The parameters $a$, $b$, $c$, $d$, $I_0$, and $\tau_{DL}$ are related to physicochemical quantities[29]. Detailed expressions are provided in Supplementary Note 2. For bare Zn, the transition from 2D diffusion to 3D diffusion dominance begins at 24 s, with the 3D diffusion being progressive. This gradual progression indicates slow, uneven growth, leading to irregular deposition and dendrite formation. In contrast, the LAPO@Zn system exhibits a rapid transition from 2D to 3D diffusion dominance at just 1.7 s, with the 3D diffusion process being instantaneous. This rapid growth highlights the effectiveness of the LAPONITE coating in mitigating ion flux vortices, resulting in more uniform nucleation and controlled deposition. Consequently, LAPO@Zn reduces the tendency for dendrite formation and growth. Although the simulations confirm improved $Zn^{2+}$ flux uniformity, direct structural characterization is necessary to validate the morphological consequences of vortex suppression.

## Zn deposition orientation

The Zn deposition on LAPO@Zn versus bare Zn exhibits notable differences in morphology and crystallographic orientation, which are key to achieving durable Zn negative electrodes. SEM images (Fig. 3a, b and Supplementary Fig. 14) indicate that Zn deposits on LAPO@Zn are predominantly vertically aligned, while those on bare Zn appear

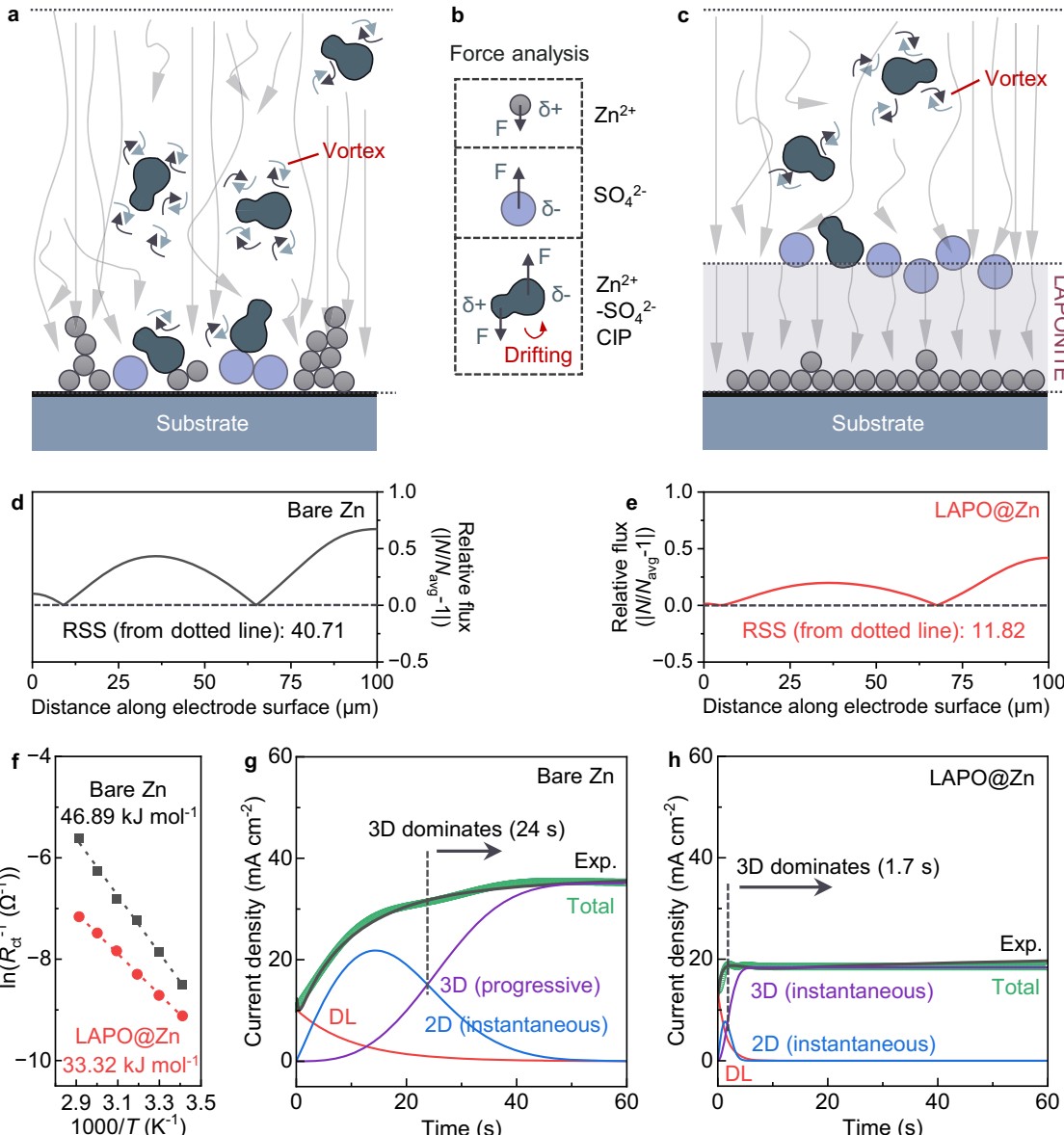

**Fig. 2 | Diffusion behavior on different electrodes. a, c** Schematic streamlines upon deposition substrates without (**a**) and with LAPONITE (**c**). **b** Force analysis of $Zn^{2+}$, $SO_4^{2-}$, and $Zn^{2+}$-$SO_4^{2-}$ CIP under applied electric field. **d, e** Simulated $Zn^{2+}$ flux density on bare Zn (**d**) and LAPO@Zn (**e**) electrode surfaces. $N$ and $N_{avg}$ represent the local and average $Zn^{2+}$ flux densities, respectively. The y-axis shows the absolute deviation of normalized flux ($|N/N_{avg}-1|$). RSS (residual sum of squares) was calculated by squaring each deviation and summing across the surface. Data collected from ionic concentration field simulations in Supplementary Fig. 7. **f** Arrhenius plots of $R_{ct}^{-1}$ values at different temperatures (from 20 °C to 70 °C). The activation energies for the interfacial charge transfer processes were determined by calculating the slope of the fitted line. **g, h** Chronoamperograms (CAs) of bare Zn (**g**) and LAPO@Zn (**h**) electrodes at an overpotential of −150 mV. DL, 2D, and 3D are double-layer, two-dimensional, and three-dimensional currents. Source data are provided as a Source Data file.

disordered and randomly stacked. Grazing incidence X-ray diffraction (GIXRD) analysis further confirms this (Fig. 3c), showing a lower intensity ratio of $I_{(002)}$ to $I_{(100)}$ for LAPO@Zn compared to bare Zn, indicating preferential growth along the (100) plane, and the Zn layer on LAPO@Zn maintains a low $I_{(002)}/I_{(100)}$ ratio even after long-term cycling (Supplementary Fig. 15). This alignment is attributed to the LAPONITE coating, which channels $Zn^{2+}$ through its separated charge centers and lamellar structure, promoting controlled and uniform growth, and is further supported by the more stabilized interfacial impedance evolution during stepwise Zn deposition, as revealed by operando EIS (Supplementary Figs. 16 and 17).

In-situ X-ray computed tomography (X-ray CT) inspection provides further insight into the deposition orientation. After substantial Zn deposition on bare Zn and LAPO@Zn substrates, we collected their corresponding X-ray CT images and performed segmentation of all Zn deposits above the original substrates (Supplementary Fig. 18). Over 3,000 units were identified from each electrode and mapped onto a polar axial system (Fig. 3d, e). The orientation of each Zn deposit was characterized by two angles: the in-plane angle (ranging from 0 to 360 degrees) and the out-of-plane angle (ranging from 0 to 90 degrees; angles beyond 180 degrees are physically irrelevant due to the upward growth direction of Zn, and 135 degree is equivalent to 45 degrees owing to axial symmetry). We primarily focus on the out-of-plane angle to determine whether the Zn deposits are vertically, horizontally, or obliquely stacked. As shown in Fig. 3d, the out-of-plane direction angle distribution of Zn deposits on bare Zn is nearly uniform across all radial directions, indicating randomly oriented growth and a lack of spatial control. In contrast, Fig. 3e reveals that Zn deposits on

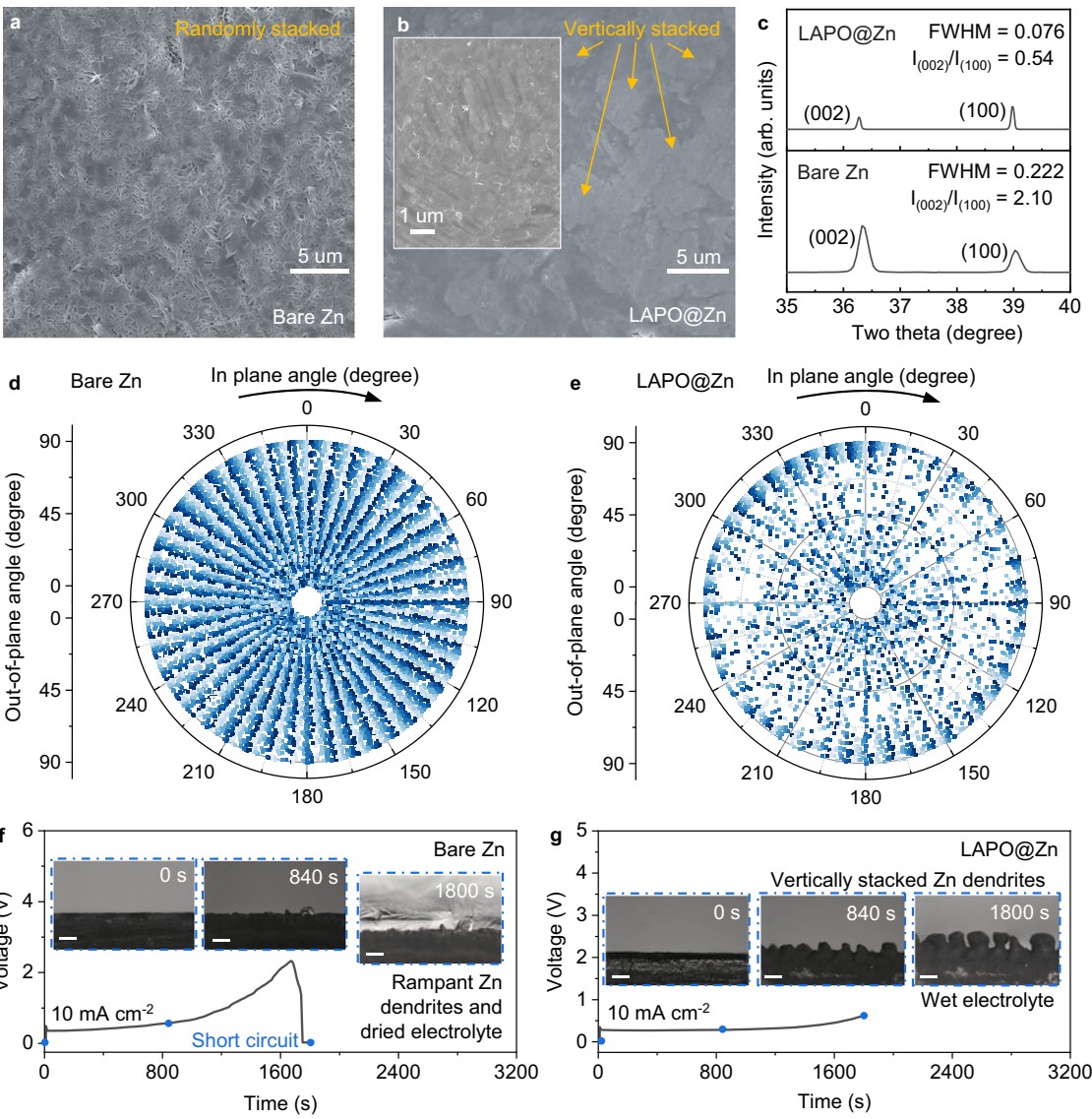

**Fig. 3 | Zn deposition orientation on different electrodes. a, b** SEM images of the Zn foils cycled with bare Zn (**a**) and LAPO@Zn (**b**) after 1 h plating at 1 mA cm$^{-2}$. **c** Grazing incidence X-ray diffraction (GIXRD) patterns of bare Zn and LAPO@Zn after 1 h plating at 1 mA cm$^{-2}$. FWHM represents the full width at half maximum of the peak. **d, e** Statistical analysis of the in-plane and out-of-plane orientation of the particles on bare Zn (**d**) and LAPO@Zn (**e**) after cycling in an in-situ X-ray CT testing (relative 3D rendered images are shown in Supplementary Fig. 18). **f, g** In-situ optical microscopy and voltage profiles of bare Zn (**f**) and LAPO@Zn (**g**) at 10 mA cm$^{-2}$. The inset pictures show the corresponding optical images with scale bars of 200 μm. Source data are provided as a Source Data file.

LAPO@Zn are predominantly oriented near 0 and 90 degrees, forming a bimodal distribution corresponding to the smallest and largest circles in the polar plot. This reflects a more ordered and vertically aligned deposition pattern. The penetrative X-ray CT analysis offers critical insight into how LAPONITE's layered channels and charge-separated structure guide Zn$^{2+}$, favoring growth along the (100) plane, which typically aligns at an angle of 70–90 degrees relative to the substrate (i.e., more vertically stacked). Under higher current densities (10 mA cm$^{-2}$), the difference becomes even more apparent (Fig. 3f, g). Bare Zn shows rampant and disordered dendritic growth, leading to electrolyte drying and short-circuiting (Fig. 3f). In contrast, LAPO@Zn promotes vertically aligned, stable growth while preserving electrolyte integrity (Fig. 3g). This underscores the ability of LAPO@Zn to sustain controlled Zn deposition, even at high current densities.

These results confirm that LAPONITE mitigates Zn$^{2+}$ vortex formation, aligning Zn growth. This enhanced control reduces dendrite formation, especially the dendrite volume, branch size and curvature, contributing to improved negative electrode uniformity and safety,

which are essential for high-performance metal batteries. Beyond growth orientation, the compactness and grain structure of Zn deposits also play a comparably vital role in dendrite suppression and long-term stability.

## Zn deposition compactness

Beyond orientation, GIXRD analysis reveals that the full width at half maximum (FWHM) of the Zn peak for LAPO@Zn is much narrower (0.076°) compared to bare Zn (0.222°) (Fig. 3c), indicating larger grain sizes and enhanced crystallinity in the Zn deposits on LAPO@Zn. This contributes to a more compact structure. Curvature analysis from X-ray CT segmentation (Fig. 4a, b) further shows that the average curvature of Zn deposits on bare Zn is 0.019, whereas on LAPO@Zn it is reduced to 0.0097. This reduction points to a smoother surface on LAPO@Zn, which helps prevent localized current density hotspots and thus reduces the risk of dendrite formation. The radius distribution analysis of Zn deposits (Fig. 4c, d) corroborates these observations, with the mean radius of Zn on bare Zn at 2.60 μm, compared to

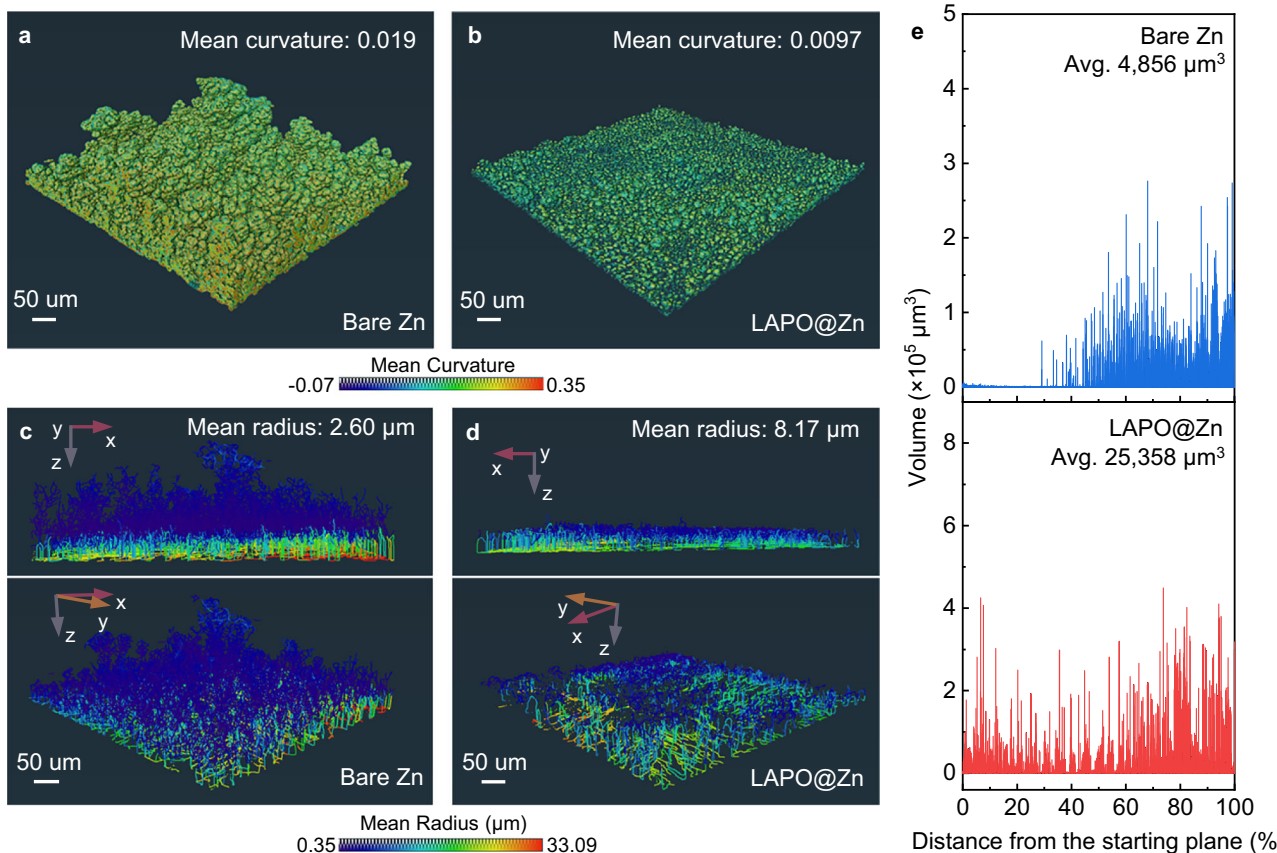

**Fig. 4 | X-ray CT imaging of Zn depositions on different electrodes. a–d** Regions of interest (ROI) demonstrating the mean curvature distribution of bare Zn (**a**) and LAPO@Zn (**b**), and the mean radius of bare Zn (**c**) and LAPO@Zn (**d**) after in-situ X-ray CT testing. **e** Zn segmented volume variation in bare Zn and LAPO@Zn in the regions corresponding to **a-d**. Source data are provided as a Source Data file.

8.17 μm on LAPO@Zn. A larger radius implies a coarser grain, more uniform deposition, minimizing dendritic growth and fostering a more robust morphology.

Furthermore, volume distribution histograms (Fig. 4e) illustrate the advantages of LAPONITE. Although the morphology of bare Zn appears to cover a larger spatial extent (Fig. 4c), this growth mainly consists of thin, elongated structures, leading to a smaller average segmentation volume compared to LAPO@Zn. The mean volume of Zn deposits on LAPO@Zn is substantially larger (25,358 μm³) compared to bare Zn (4,856 μm³). This discrepancy arises because LAPO@Zn facilitates the formation of more impact and coarser deposits, resulting in a seemingly smaller spatial extent but larger-size segments (Fig. 4d), unlike the fine, needle-like dendrites found in bare Zn. Collectively, the combination of reduced curvature, larger radii, and increased segmentation volume highlights LAPONITE's capability to mitigate $Zn^{2+}$ vortices, enabling smoother, denser, and more durable Zn growth. This enhancement is crucial for achieving high-performance, long-lasting Zn negative electrodes.

Beyond enhancing Zn deposit compactness, LAPONITE also reduces ZHS formation by separating $Zn^{2+}$ from $SO_4^{2-}$, which inhibits $SO_4^{2-}$ from reacting with hydrogen evolution reaction (HER)-derived $OH^-$. Overlapped segmentation images (Supplementary Fig. 19) show that LAPONITE acts as a barrier, blocking ZHS from forming on the Zn substrate. Additionally, post-cycling analysis exhibits a considerably weaker ZHS peak on the LAPO@Zn surface compared to the pronounced peak on bare Zn, consistent with DRT analysis from in-situ EIS tests (Supplementary Figs. 20, 21 and Supplementary Table 2). Complementary operando X-ray radiography (Supplementary Fig. 22) shows that LAPO@Zn maintains a stable and homogeneous interface with reduced ZHS generation, whereas bare Zn exhibits uneven bulges

and signs of corrosion. These findings indicate that LAPONITE alleviates the increase in charge transfer resistance ($R_{ct}$) associated with ZHS accumulation, leading to a more stable interface compared to bare Zn. To validate the benefits of compact and uniform Zn deposits, electrochemical performance was evaluated in both symmetric cells and full cells.

## Electrochemical performance

The cycling performance of Zn negative electrodes and Zn ‖MnO₂ full cells was studied to assess the impact of the LAPONITE coating. At a current density of 1 mA cm⁻², the LAPO@Zn symmetric cell cycled stably for over 1,200 h, whereas the bare Zn symmetric cell began to fail after only 94 h (Fig. 5a). Notably, the LAPO@Zn symmetric cell also maintained stable cycling at a high current density of 5 mA cm⁻² (Supplementary Fig. 23). Inductively coupled plasma optical emission spectroscopy (ICP-OES) analysis further confirmed minimal LAPONITE dissolution after cycling, demonstrating its long-term interfacial stability (Supplementary Table 3). During continuous plating/stripping, Zn deposits form exclusively beneath the coating, with no accumulation detected within or atop the LAPONITE layer (Supplementary Fig. 24), corresponding to its long-term effectiveness. In Zn‖Cu asymmetric cells, LAPO@Cu exhibited stable cycling for more than 300 h with an average Coulombic efficiency (CE) of 97.4%, compared to bare Cu, which failed after 41 h with an average CE of 92.9% (Supplementary Fig. 25). These results manifest how LAPONITE mitigates $Zn^{2+}$ vortex formation, resulting in uniform Zn deposition, enhanced reversibility of Zn plating/stripping on Cu foil, and minimized side reactions towards the electrolyte. To rule out the influence of CMC itself, control experiments using only CMC-coated Zn electrodes were conducted and showed no comparable performance enhancement

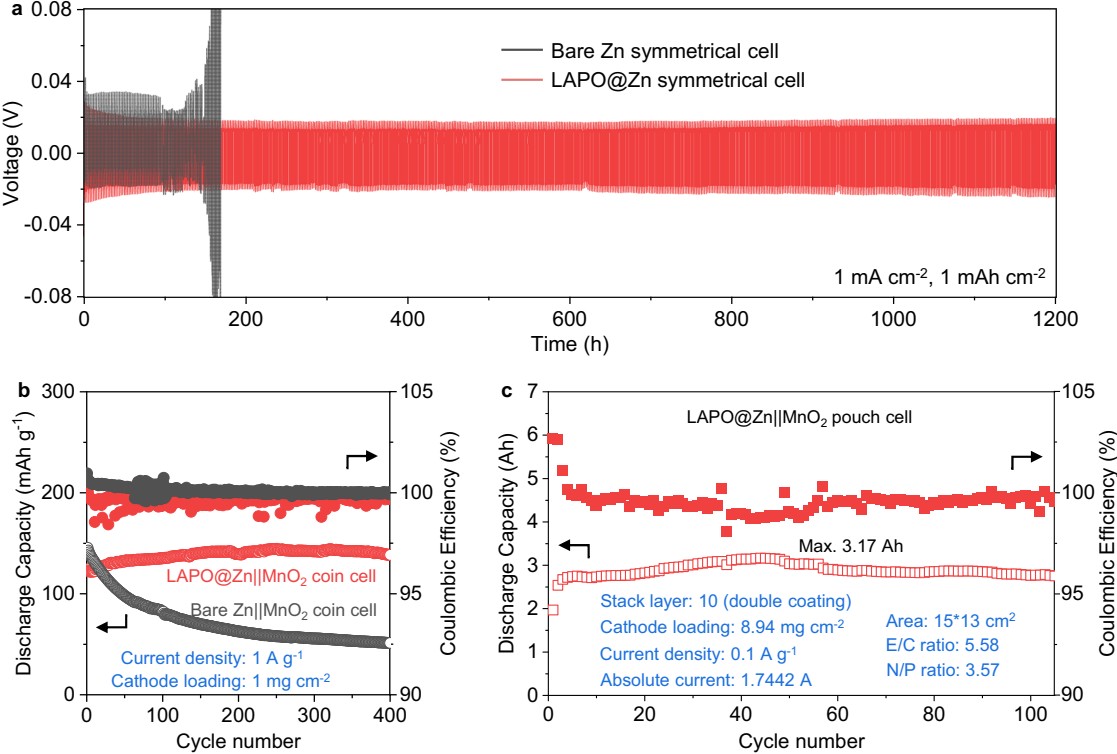

**Fig. 5 | Electrochemical performance of cells with different electrodes.**
a Galvanostatic cycling curves of Zn ||Zn symmetric cells with bare Zn and
LAPO@Zn electrodes at 1 mA cm⁻² with a fixed capacity of 1 mAh cm⁻².

b Galvanostatic cycling performance of Zn ||MnO₂ coin cells at 1 A g⁻¹.
c Galvanostatic cycling performance of the LAPO@Zn ||MnO₂ pouch cell at 0.1 A g⁻¹.
Source data are provided as a Source Data file.

(Supplementary Figs. 26-28). Transitioning to full-cell configurations, the cyclic voltammetry (CV) curves of bare Zn ||MnO₂ and LAPO@Zn ||MnO₂ (Supplementary Fig. 29) exhibited no significant differences, indicating that the LAPONITE interlayer remains stable during full cell cycling without affecting the positive electrode's reaction mechanism. In addition, long-term cycling at 1 A g⁻¹ demonstrated the superior stability of LAPO@Zn ||MnO₂, which showed no noticeable capacity fading over 400 cycles, while bare Zn ||MnO₂ experienced rapid capacity fading, retaining only 80% of its initial capacity within 25 cycles and decreasing to 35.2% by 400 cycles (Fig. 5b). Additionally, LAPO@Zn ||MnO₂ maintained stability for over 3,500 cycles at 2 A g⁻¹, demonstrating excellent durability under high-rate conditions (Supplementary Fig. 30). To assess scalability, Zn ||MnO₂ pouch cells were assembled using commercially available materials without complex treatments. As shown in Fig. 5c, the LAPO@Zn ||MnO₂ pouch cell (optical photograph shown in Supplementary Fig. 31), featuring the stabilized LAPO@Zn negative electrode, delivered a maximum capacity of 3.17 Ah during 100 cycles, even under harsh conditions, including an electrolyte amount to positive electrode capacity (E/C) ratio of 5.58 g Ah⁻¹, a capacity ratio of the negative electrode to the positive electrode (N/P) of 3.57, and a low current density of 0.1 A g⁻¹. The LAPO@Zn ||MnO₂ pouch cell greatly outperformed the bare Zn ||MnO₂ pouch cell that decayed rapidly within the initial 20 cycles (Supplementary Fig. 32). Higher-capacity cells often suffer from faster capacity fading, a typical trade-off observed in previously reported aqueous zinc-ion pouch cells (Supplementary Table 4). Remarkably, the LAPO@Zn ||MnO₂ pouch cell in this work maintained both high capacity and long cycle life, indicating improved promise for practical, rechargeable Zn-ion batteries with commercial viability. Post-mortem analysis revealed that the degradation of the LAPO@Zn ||MnO₂ pouch cell primarily originated from the positive electrode side (Supplementary Figs. 33–35), further demonstrating the stability of the LAPONITE-coated Zn negative electrode.

## Discussion

This study underscores the critical role of mitigating $Zn^{2+}$ flux vortices in helping achieve stable and efficient Zn negative electrodes for aqueous Zn-ion batteries. By designing and deploying a LAPONITE coating with distinct charge centers, Zn deposition was effectively directed, leading to uniform, dense, and vertically aligned growth. In-situ X-ray CT revealed that this strategy not only guided Zn deposition along the (100) plane but also significantly enhanced the compactness of the deposits, resulting in smoother and less dendritic structures. Consequently, LAPO@Zn ||MnO₂ pouch cells exhibited superior cycling performance, achieving a peak capacity of 3.17 Ah at cycle 44 and maintaining stable operation over 100 cycles under harsh conditions, in contrast to the rapid degradation experienced in bare Zn || MnO₂ cells. These findings offer a potential pathway to commercially viable, rechargeable Zn-ion batteries, bridging advanced materials design with practical, scalable energy storage solutions.

## Methods

### Electrolyte and electrode preparation

The 2 M $ZnSO_4$ electrolyte was obtained by dissolving 0.1 mol of $ZnSO_4 \cdot 7H_2O$ ( > 99%, VWR chemicals) in deionized water (DI water) and adjusting the final volume to 50 mL with magnetic stirring for 15 min. Zinc foils (99.9%, Sigma-Aldrich) coated with LAPONITE (BYK Additives Ltd., used as received) interlayers were prepared by dissolving LAPO-NITE and carboxymethyl cellulose sodium (CMC, Sigma-Aldrich) in an appropriate amount of DI water at a mass ratio of 1:1, followed by casting the resulting slurry onto the Zn foil. The mass loading of LAPONITE was set at 1 mg cm⁻² (as identified to be the optimal loading in Supplementary Note 3). After drying in a vacuum oven, LAPO@Zn was cut into round pieces of 16 mm diameter for use. The LAPO@Cu electrode was prepared in a similar manner, using Cu foil (25 μm, MTI Corp) in replace of Zn foil. The MnO₂ positive electrodes were prepared by manually mixing 70 wt.% commercial MnO₂ powder

(Sigma-Aldrich), 20 wt.% acetylene black (battery grade, MTI), and 10 wt.% poly(vinylidene fluoride) (average Mw ~534,000) in N-methyl-2-pyrrolidine (>99%, Sigma-Aldrich) using a mortar with pestle at room temperature for 30 min. The slurry was then cast onto the current collectors using a doctor blade, dried under vacuum at 60 °C for 12 h, and punched into round electrodes of 16 mm diameter using a disc cutter. No calendaring or additional heat treatment was performed before transferring the electrodes into the Ar-filled glovebox for storage. Hydrophilic type carbon paper (TORAY) and high-purity stainless steel foil (50 μm, Zhejiang Vastech Co., Ltd) were served as positive electrode current collectors in coin cells and pouch cells, respectively.

## Material characterizations

SEM was conducted on JEOL-JSM-6700F (Voltage: 5 kV, emission current: 110.8 μA). The in-situ optical microscope was performed on a VisiScope® BL254 T1 (VWR) instrument with a purpose-designed electrolytic cell in a symmetric configuration of Zn ‖Zn. The GIXRD patterns of Zn foils were obtained on a PANalytical Empyrean device (Cu Kα radiation; 40 kV, 40 mA) with a step size of 0.05° and a scan rate of 0.1° s$^{-1}$. The chemical states were investigated by XPS (Kratos Axis SUPRA). Fourier transform infrared spectroscopy (FTIR) spectra were collected by an Attenuated Total Reflectance Fourier transform infrared spectroscopy (BRUKER, platinum-ATR). ICP-OES (JY/T015-1996) was used to analyze the sample composition quantitatively. Contact angles were acquired on Kruss DSA25E (Germany). Zeta potential measurements were carried out using a Malvern Zetasizer Nano ZS. TOF-SIMS measurements were conducted with a PHI nano TOF 3. A Bi$_3^+$$^+$ beam (30 kV, 2 nA) with raster size of 100 μm was used as the primary beam to detect the samples. An Ar$^+$ beam (2 kV, 100 nA, with a sputtering area of 400 μm × 400 μm) was applied for depth profiling analysis. The sputtering rate is around 9.16 nm min$^{-1}$ on SiO$_2$. Bruker's Dimension Icon atomic force microscope was used to record images of AFM and KPFM, which with scanning frequency of 1 Hz and area of 3 μm.

## Electrochemical measurements

The Zn ‖Zn symmetric cells, Zn ‖Cu cells, Zn ‖MnO$_2$ coin cells and Zn ‖ MnO$_2$ pouch cells were assembled to evaluate the electrochemical performances. All coin-cell tests were conducted using CR2032-type coin cells on a Neware battery test system (Shenzhen, China). The Zn ‖ Cu cells were tested with a cut-off potential of 0.5 V during charging. A single layer of 19 mm-diameter glass fiber separator (Whatman GF/D) was used in coin cells. The amounts of electrolytes for each coin cell were controlled as 100 μL. The Zn negative electrode had a thickness of 70 μm and a diameter of 16 mm. The mass loading of MnO$_2$ used in coin cells and pouch cells were 1 mg cm$^{-2}$ and 4.47 mg cm$^{-2}$ (8.94 mg cm$^{-2}$ for one single positive electrode slice as it was double coated), respectively. The cycling voltage windows for Zn ‖MnO$_2$ coin cells and pouch cells are 0.8–1.8 V and 0.8–1.9 V, respectively. The pouch cell used 15.3 g electrolyte, corresponding to an E/C ratio of 5.58 g Ah$^{-1}$, calculated based on the fifth-cycle (initial reversible) capacity of 2.7419 Ah. Both the positive and negative electrodes in pouch cells were rectangular sheets measuring 15 cm × 13 cm, and 10 layers of 150 μm thick glass fiber separator (Chongqing Ouleji Co., Ltd) were used throughout. A Biologic VMP-3 electrochemical workstation was used to conduct in-situ EIS investigation, CA of Zn ‖Zn symmetric cells at an overpotential of −150 mV, linear polarization curves at 10 mV s$^{-1}$ (with Zn foil as the working electrode, Pt as the counter electrode and Ag/AgCl as the reference electrode), and linear sweep voltammetry (LSV) curves at 5 mV s$^{-1}$ (using Zn ‖Ti half cells). Temperature-dependent potentiostatic EIS measurements were conducted on a AutoLab PGSTAT302N system. A 10-mV sinusoidal voltage signal was applied under potentiostatic control after the system reached open-circuit voltage (OCV) and stabilized. The frequency was swept logarithmically from 100 kHz to 10 mHz with 10 points per decade. All tests were conducted under ambient conditions (25 °C, air-conditioned room). All electrochemical measurements were repeated using three independently prepared Zn ‖Zn symmetric cells and Zn ‖MnO$_2$ coin cells, and consistent performance was observed across the replicates (detailed in Supplementary Note 4). The representative galvanostatic voltage profiles of Zn ‖MnO$_2$ coin cells and pouch cells at various cycles are provided in Supplementary Note 5.

## Theoretical calculation methods and models

All first principle simulations were performed by VASP (Vienna Ab initio Simulation Package) code[30]. All structures were obtained from the Materials Project by the comprehensive analysis between the experiment and simulation[31]. The generalized gradient framework and projector augmented wave (PAW) method were adopted with a periodic slab model using the projected augmented wave approach to describe the electron–ion interactions with a cut-off energy of 450 eV[30,32,33]. Brillouin zone integration was accomplished using a 3 × 3 × 1 Monkhorst–Pack k-point mesh[34]. To illustrate the long-range dispersion interactions between the adsorbates and catalysts, we employed the D3 correction method of Grimme et al.[35]. The vacuum layer was set as 20 Å. VESTA was adopted to visualize the electron localization functions of the surface property[36]. For the electron localization function analysis, to fully reveal the enrichment of the electrons, we first perform the geometry optimization of the 2 × 2 × 1 slab, and then perform the electronic simulation at the 20 × 15 × 1 supercell. All vortices flow finite element analysis was performed by COMSOL Multiphsics. All structures and materials were constructed by comprehensive analysis.

## Computational fluid dynamics simulation

The solution domain of the model is shown in Supplementary Fig. 5, taking the 100 μm × 100 μm 2D region above the electrode, with the blue part ($\Omega_1$) as the electrolyte and the gray part ($\Omega_2$) as the porous film. In this model, the fluid motion is governed by the Navier-Stokes equations for incompressible fluids, and the effect of the porous film is described by Darcy's law, which is present in the steady state case:

$$\begin{cases} \nabla \cdot u = 0 \\ \frac{1}{\epsilon^2}\rho(u\nabla)u = \nabla \cdot (-pI + K) + F_{\text{porous}} \end{cases} \tag{5}$$

where the porosity $\epsilon$, the shear stress K in the two regions take the values of:

$$\epsilon = \begin{cases} 1\,(in\,\Omega_1) \\ 0.3\,(in\,\Omega_2) \end{cases} \tag{6}$$

$$K = \begin{cases} \mu\left(\nabla u + (\nabla u)^T\right)(in\,\Omega_1) \\ \frac{1}{\epsilon}(\mu\left(\nabla u + (\nabla u)^T\right) - \frac{2}{3}\mu(\nabla u)I)\,(in\,\Omega_2) \end{cases} \tag{7}$$

The Kozeny-Carman model is introduced for the calculation of the additional force $F_{\text{porous}}$ on the fluid by the porous medium:

$$F_{\text{porous}} = \begin{cases} 0\,(in\,\Omega_1) \\ \left(\frac{180\mu(1-\epsilon)^2}{d_p^2\epsilon^3} + \frac{\rho}{\epsilon^2}\nabla u\right)u\,(in\,\Omega_2) \end{cases} \tag{8}$$

$d_p = 1\,nm$ is the nano-channel diameter. At the same time, the ion concentration is controlled by the convection-diffusion equation, i.e., the:

$$\begin{cases} -D_1\nabla^2 c_{\text{free}} + u \cdot \nabla c_{\text{free}} = R \\ -D_2\nabla^2 c_{\text{abso}} + u \cdot \nabla c_{\text{abso}} = R \end{cases} \tag{9}$$

Based on the assumption of electroneutrality, for example, the electrolyte is a dilute solution of zinc sulfate, the concentration of zinc ions is equal to the concentration of sulfate ions, denoted as $c_{\text{free}}$, and there is adsorption of ions in a porous medium, and we denote the concentration of the adsorbed ions as $c_{\text{abso}}$. $D$ is the diffusion coefficient, and in the electrolyte, $D_1 = D_L I$, $D_L = 0.8 \times 10^{-7} m^2/s$; in the control film, the $D_1 = \begin{pmatrix} D_L \epsilon & 0 \\ 0 & 5 D_L \epsilon \end{pmatrix}$; in the LAPONITE film,

$$D_1 = \begin{pmatrix} D_L \epsilon (1 - \frac{c_{\text{abso}}}{c_{\text{eq}}}) & 0 \\ 0 & 5 D_L \epsilon (1 - \frac{c_{\text{abso}}}{c_{\text{eq}}}) \end{pmatrix},$$ where $c_{\text{eq}}$ is the equilibrium

concentration. $D_2$ is close to 0, and it is worthwhile to take $10^{-11} m^2/s$ for calculation. Absorbance $R$ is 0 in the control film and $R = R_0(c_{\text{free}} - c_{\text{abso}})$ in the LAPONITE film, with $R_0$ being a positive real number. As for boundary conditions, we model the ion transport of the fluid above the electrodes by constructing artificial vortices by setting the velocity at the boundary above the computational domain to a nonzero real number of appropriate sizes, and the velocity at the other boundaries of the computational domain to 0. $c_{\text{free}}$ is the initial concentration at the boundary above the computational domain, 0 at the boundary below (the electrodes), and the concentration at the rest of the boundaries satisfies $n \cdot \nabla c_{\text{free}} = n \cdot \nabla c_{\text{abso}} = 0$. Other details for CFD simulations are shown in Supplementary Note 6.

### Synchrotron X-ray imaging and image processing

A polychromatic X-ray beam with a peak photon energy at 75 keV used for X-ray radiography and tomography acquisition at the beamline of ID19 in the ESRF. A 50 μm thick single crystal scintillator, cerium-doped (LuAG:Ce, Czech Republic) was used. The sample-to-detector distance was 80 mm to achieve some propagation-based phase-contrast effect. The field of view (FOV) was 2160 × 2560 pixels with an effective pixel size of 0.7 μm via a 10× microscope objective lens. Operando X-ray radiography was performed to capture the dynamic plating process until the cell short-circuit. The exposure time was 0.05 s. In addition, synchrotron X-ray microtomography was carried out before and after electrodepositions in order to capture volumetric changes in the sample. For tomography, 4000 projections were taken at 300 ms per projection through continuous rotation of 360°. The reconstruction was carried out using standard FBP with ESRF in-house software (NABU reconstruction package[37]). The image processing was performed using the Avizo and Fiji to visualize and quantify the dendrites. The orientation analysis was done through the Auto Skeleton module.

## Data availability

All data that support the findings of this study are presented in the Manuscript and Supplementary Information, or are available from the corresponding author upon request. X-ray data[38] Source data are provided with this paper.

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

## Acknowledgements

We acknowledge Beamline ID19 at The European Synchrotron Radiation Facility (ESRF) for the experiments (MA-6153, https://doi.org/10.15151/ESRF-ES-1671359888, W.D.). B.L. and P.P.P. would like to acknowledge funding by the Engineering and Physical Sciences Research Council (EPSRC) for the International Centre to Centre Collaboration with the ESRF grant reference EP/W003333/1 and the Henry Royce Institute established through EPSRC grants EP/R00661X/1, EP/P025498/1 and EP/P025021/1. This work was also supported by the EPSRC (EP/V027433/3, EP/L015862/1, G.H.), UK Research and Innovation (UKRI) under the UK government's Horizon Europe funding guarantee (101077226; EP/Y008707/1, G.H.), Faraday Institution (EP/S003053/1) DEFINE project (FIRG050, G.H.), ReLiB project (FIRG057, P.R.S and W.D.) and State Key Laboratory of Advanced Technology for Materials Synthesis and Processing (Wuhan University of Technology, L.M. and G.H.). The authors would like to acknowledge Zhejiang Vastech Co., Ltd. for their technical support in the pouch cell assembling process.

## Author contributions

G.H. and Y.D. conceived and designed the research. W.D., P.P.P., B.L., A.R., Y.D., and P.R.S. conducted the in-situ X-ray radiography and X-ray CT characterizations, as well as related data processing. Y.D. and X.G. wrote the manuscript with guidance from L.M. and G.H. H.D. and X.G. carried out CFD simulations. C.Z. conducted DFT calculations. Y.D., X.G., C.S., C.Y., J.H.L., W.Z., J.W.L. and Y.L. performed electrochemical tests and analyzed the data. All authors discussed and commented on the manuscript.

## Competing interests

The authors declare no competing interests.
