## [Transparent Peer Review file · Nature Communications]

Mitigating ion flux vortex enables reversible zinc electrodeposition

Corresponding Author: Professor Guanjie He

Version 0:

Reviewer comments:

Reviewer #1

(Remarks to the Author)

The author conducted research on metal anodes, specifically investigating the suppression of dendrite growth in Zn anodes by using LAPONITE coatings. The study employed in-situ X-ray computed tomography and computational fluid dynamics simulations to demonstrate that the LAPONITE coating suppresses ionic vortex formation and guides uniform, dense, vertically aligned Zn growth along the (100) plane. This approach significantly mitigates dendrite formation and leads to stable performance in a Zn-MnO₂ pouch cell over 100 cycles. However, minor revisions are required for publication in the Nature Communications.

1. Please check whether the charge density difference results in Figure S2 for Zn²⁺ and SO₄²⁻ adsorption on the edge and surface are plotted using the same isosurface value. Additionally, please label the isosurface value. Additionally, it is necessary to calculate the adsorption energy of Zn²⁺ and SO₄²⁻. This will allow for a more precise confirmation of charge separation occurring at the edge and surface. Lastly, calculating the formation energy of Zn ions when entering the LAPONITE structure would help demonstrate the possibility of ion incorporation, providing insight into whether ions can actually enter the coating layer.
2. Please check whether the scale bar range in the ELF results obtained from DFT calculations (Figure S10) is the same. ELF is calculated within a range of 0 to 1 to analyze ionic or covalent bonding characteristics, and it also provides insights into local electron distribution properties. Verify whether Figures S10a and S10b are plotted using the same scale from 0 to 1, label the scale bar with numerical values, and if they were plotted with different scales, please adjust them to the same scale.
3. Figure 2b is mentioned before Figure 2a. The figure order or the content of the paper may need to be revised.

Reviewer #2

(Remarks to the Author)

The article presents a Zn-MnO₂ pouch cell featuring LAPONITE-coated Zn anodes. Through advanced in-situ characterization, the authors offer valuable insights into the internal structural evolution and deposition orientation, providing a promising approach for the development of scalable, high-performance metal-anode batteries. While the study holds promise, there are few areas that require improvement before considering it for publication.

The main issues with the article are as follows:

1. The zinc anode coating is prepared with CMC and LAPONITE in a 1:1 ratio. Since CMC accounts for a significant proportion, how can the influence of CMC on the uniform deposition of zinc ions and the suppression of water molecules be excluded?
2. During continuous plating/stripping, do zinc ions deposit not only beneath the coating, but also in the middle and on top of the coating?
3. According to Fig. 1b, what drives the migration of zinc ions through the bending path between LAPONITE layers? Does the bending migration path affect the zinc ion migration, leading to deposition between the layers?
4. How were the RSS data in Fig. 2d and e obtained?
5. The temperature-dependent EIS data in Fig. 2f is too limited, and the activation energy fitting is not precise enough.

6. Line 195, the significance of the physical and chemical parameters such as a , b , $c3DI$, $c3DP$, $d3DI$, $d3DP$, I_0 , τDL is unclear.
7. Please provide the XRD data of the original zinc foil, the gradient process of the 100 crystal plane dominant orientation after LAPO@Zn cycling, and the XRD data for the 002, 100, 101, and other crystal planes.
8. The SEM images in Fig. 3b are not clear enough, making it difficult to identify the dominant orientation of the 100 crystal plane.
9. The description of deposition orientation in Fig. 3d and e is not clear enough.
10. Mineral materials with distinct positive and negative charge centers are not limited to LAPONITE. Do other clay materials such as attapulgite, kaolinite, and montmorillonite exhibit similar effects?
11. The microscopic morphology and macroscopic features (such as thickness) of the coating are not clear enough. Additionally, does the coating thickness and proportion have an impact?
12. There are formatting errors in the citation of ref2 and ref20 in the introduction.
13. Supplementary Fig. 24-27 are not described in the main text.

By addressing these concerns, the paper can be more suitable for publication.

Reviewer #3

(Remarks to the Author)

This manuscript presents significant contributions to Zn anode stabilization and dendrite suppression. The scientific rationale is sound, and the experimental techniques are advanced. However, major improvements are required in the clarity of writing, data interpretation, and additional experimental validation of theoretical claims. Addressing these points will strengthen the manuscript's impact and relevance.

- 1) The anode surface modification was done by a mixture of LAPONITE and CMC, but the manuscript only discusses LAPONITE. It is well-known that CMC is also efficient in preventing dendrite formation and parasitic side reactions. The surface layer thickness is not provided.
- 2) The manuscript presents compelling experimental data but lacks clear, concise transitions between sections. The introduction should better structure the challenges of Zn anodes and why ion flux vortices specifically affect deposition morphology.
- 3) Ion flux vortex: This term is introduced without sufficient context. Has it been previously reported in electrochemical literature, or is this a new concept introduced by the authors? Clarification is needed.
- 4) (This translates into a 3.54-Ah Zn-MnO₂ pouch cell with stable performance over 100 cycles.) While promising, 100 cycles is relatively short for practical applications. Stable performance needs quantification—what capacity retention, Coulombic efficiency, or failure mode was observed?
- 5) While the X-CT imaging and CFD simulations provide strong evidence of ion flux vortex suppression, additional validation via operando electrochemical impedance spectroscopy (EIS) during Zn deposition would further confirm this mechanism.
- 6) While the X-CT imaging and CFD simulations provide strong evidence of ion flux vortex suppression, additional validation via operando electrochemical impedance spectroscopy (EIS) during Zn deposition would further confirm this mechanism.
- 7) The DFT calculations supporting LAPONITE's charge-separation effects are insightful, but no clear experimental validation of charge redistribution is provided. Techniques such as X-ray photoelectron spectroscopy (XPS) core-level shift analysis or zeta potential measurements could strengthen this claim.
- 8) A. The pouch cell cycling performance is promising, but the long-term degradation mechanisms remain unclear. Post-mortem analysis of cycled anodes (e.g., SEM/EDS mapping after 100 cycles) would be beneficial. B. The pouch cell cycling performance is promising, but the long-term degradation mechanisms remain unclear. Post-mortem analysis of cycled anodes (e.g., SEM/EDS mapping after 100 cycles) would be beneficial. Suggestion: Provide post-cycling characterization of the Zn anode and discuss the advantages of LAPONITE compared to other anode modification strategies.
- 9) Figure 1: The color scheme for Zn and SO₄²⁻ separation should be more distinct to enhance readability. The schematic in Fig. 1b should include charge labels to clarify Zn²⁺ and SO₄²⁻ behavior.
- 10) Figure 2: CFD streamline plots (Fig. 2a,c) lack a scale bar or color legend to indicate flow intensity. Clarify what simulation parameters were used in CFD to avoid ambiguity.
- 11) Figure S22: Zn||MnO₂ or LAPO@ Zn||MnO₂ which one is correct?

The manuscript can be considered for publication in Nature or a similarly high-impact journal upon addressing these revisions.

Version 1:

Reviewer comments:

Reviewer #1

(Remarks to the Author)

The overall manuscript has been revised in the desired direction, and I believe it is now suitable for publication in Nature

Communications.

Reviewer #2

(Remarks to the Author)

The authors have adequately addressed all the concerns raised in the previous review. The additional experiments and revised explanations significantly strengthen the manuscript. The new data support the conclusions, and the text revisions, such as expanded discussion in Results, improve clarity and rigor. The manuscript now meets the standards for publication.

Reviewer #3

(Remarks to the Author)

The authors replied to all my comments and I am satisfied with their answers. The revised manuscript can be accepted by Nature Communications.

Thank you.

Note: In this response letter, comments suggested by reviewers are in **blue font**, our responses are in **black font**, and the updated contents in the revised manuscript and revised Supplementary Information are **marked in red**. The figures in our revised Manuscript, revised Supplementary Information, and used in Response Letter only are displayed in the form of Fig. XX, Supplementary Fig. XX, and Fig. RXX, respectively.

Response to Reviewer #1:

General Comments:

The author conducted research on metal anodes, specifically investigating the suppression of dendrite growth in Zn anodes by using LAPONITE coatings. The study employed in-situ X-ray computed tomography and computational fluid dynamics simulations to demonstrate that the LAPONITE coating suppresses ionic vortex formation and guides uniform, dense, vertically aligned Zn growth along the (100) plane. This approach significantly mitigates dendrite formation and leads to stable performance in a Zn-MnO₂ pouch cell over 100 cycles. However, minor revisions are required for publication in the Nature Communications.

Response:

We greatly appreciate Reviewer #1's recognition of our work. We have carefully addressed the concerns raised to improve the clarity and presentation of the manuscript. We believe that Reviewer #1's valuable suggestions regarding the DFT calculations have substantially strengthened this study. Our point-to-point responses are provided as follows:

Comment 1-1:

Please check whether the charge density difference results in Figure S2 for Zn²⁺ and SO₄²⁻ adsorption on the edge and surface are plotted using the same isosurface value.

Additionally, please label the isosurface value. Additionally, it is necessary to calculate the adsorption energy of Zn^{2+} and SO_4^{2-} . This will allow for a more precise confirmation of charge separation occurring at the edge and surface. Lastly, calculating the formation energy of Zn ions when entering the LAPONITE structure would help demonstrate the possibility of ion incorporation, providing insight into whether ions can actually enter the coating layer.

Response to Comment 1-1:

We sincerely appreciate Reviewer #1's insightful comments and valuable suggestions regarding the differential charge density maps and the importance of assessing the adsorption energy for different ions individually. In response, we have provided differential charge density maps of LAPONITE adsorbing Zn^{2+} and SO_4^{2-} in the revised **Supplementary Fig. 2**. We have also calculated the adsorption energies for Zn^{2+} and SO_4^{2-} separately, along with the formation energy of Zn^{2+} incorporated into the LAPONITE structure, which further support our conclusions. The revised supplementary Figures and related analysis are as follows:

Supplementary Figure 2. Density functional theory (DFT) calculation result of LAPONITE lamella. Analysis of charge density difference to investigate the adsorption behavior of Zn^{2+} and SO_4^{2-} on distinct sites (edge or surface) of LAPONITE. The chain structure of LAPONITE consists of $[\text{SiO}_4]$ tetrahedra with Mg atoms between the chains. The blue area indicates electron loss, while the yellow area presents electron enrichment. The adsorption energy to each ion (cation) is labeled in the figure.

The primary source of the significant differential charge density difference between Zn^{2+} and SO_4^{2-} at the surface is their distinct interactions with oxygen atoms. Specifically, zinc close to the surface coordinates with the surface oxygen to form bonds with an adsorption energy of -1.22 eV, while the oxygen in the sulfate group cannot bond tightly with the surface oxygen, resulting in a lower adsorption energy of -0.21 eV. At the edge, the oxygen is already saturated by interlayer metal atoms, making the exposed metal atoms more likely to coordinate with the sulfate group (-1.45 eV) rather than zinc (-0.16 eV). In addition, the formation energy when zinc enter the LAPONITE structures is calculated to be -1.73 eV, indicating a stable configuration that promotes long-cycle stability of the battery.

Please see the new **Supplementary Fig. 2** and related analysis **marked in red** in the revised Supplementary Information (on Page 3 of the revised SI).

Comment 1-2:

Please check whether the scale bar range in the ELF results obtained from DFT calculations (Figure S10) is the same. ELF is calculated within a range of 0 to 1 to analyze ionic or covalent bonding characteristics, and it also provides insights into local electron distribution properties. Verify whether Figures S10a and S10b are plotted using the same scale from 0 to 1, label the scale bar with numerical values, and if they were plotted with different scales, please adjust them to the same scale.

Response to Comment 1-2:

We sincerely appreciate Reviewer #1's insightful observation and valuable suggestions regarding the identical ELF results in **Supplementary Fig. 10**. In response, we have carefully reperformed the ELF calculations and updated the figures by adding appropriate color scale bars to the right side of panel **b**. Please see the revised **Supplementary Fig. 10** below:

Supplementary Figure 10. DFT calculation of the electron localization function (ELF). a, On LAPONITE slabs. b, On Zn slabs.

Comment 1-3:

Figure 2b is mentioned before Figure 2a. The figure order or the content of the paper may need to be revised.

Response to Comment 1-3:

We are very thankful for Reviewer #1 pointing out the inconsistency in the figure citation order. To address this, we have revised the corresponding paragraph in the main text so that **Figure 2a** is mentioned before **Figure 2b**, in alignment with the figure panel order. Please see the revised text **marked in red** on **Page 5, Line 9** of the revised manuscript.

More importantly, the separation of Zn^{2+} from SO_4^{2-} influences the behavior of near-

surface $\text{Zn}^{2+}\text{-SO}_4^{2-}$ contact ion pair (CIP). The resulting imbalance in mass transfer near the surface induces significant ion flux vortices and disrupts the uniformity of Zn deposition sites, as illustrated conceptually in the schematic streamlines of Fig. 2a. This phenomenon is attributed to the distinct response of these neutrally but asymmetrically charged CIPs to the electric field, in contrast to Zn^{2+} alone (Fig. 2b). LAPONITE effectively reduces these $\text{Zn}^{2+}\text{-SO}_4^{2-}$ CIPs in the diffusion pathway, mitigating ion flux vortices and promoting a more uniform Zn^{2+} flux and deposition (Fig. 2c).

Response to Reviewer #2:

General Comments:

The article presents a Zn-MnO₂ pouch cell featuring LAPONITE-coated Zn anodes. Through advanced in-situ characterization, the authors offer valuable insights into the internal structural evolution and deposition orientation, providing a promising approach for the development of scalable, high-performance metal-anode batteries. While the study holds promise, there are few areas that require improvement before considering it for publication. The main issues with the article are as follows:

Response:

We sincerely thank Reviewer #2 for the positive comments. We are encouraged by the Reviewer's recognition of the value and potential of our study. We have carefully addressed all the concerns raised in the following responses and revised the manuscript accordingly to enhance its clarity and overall quality. We hope that the revised version is now more suitable for publication in *Nature Communications*.

Comment 2-1:

The zinc anode coating is prepared with CMC and LAPONITE in a 1:1 ratio. Since CMC accounts for a significant proportion, how can the influence of CMC on the uniform deposition of zinc ions and the suppression of water molecules be excluded?

Response to Comment 2-1:

We sincerely thank Reviewer #2 for raising this important question, which has greatly helped us improve the rigor and scientific clarity of our manuscript. To investigate the influence of CMC alone on Zn deposition uniformity and suppression of water-induced side reactions, we conducted a series of control experiments by coating Zn foil with CMC alone (denoted as CMC@Zn), using the same areal loading (1 mg cm⁻²) as the LAPONITE-CMC composite layer applied in LAPO@Zn.

First, the Zn deposition morphology was examined via SEM after plating 1 mAh cm⁻² of Zn at 1 mA cm⁻² (**Supplementary Fig. 32**). As shown, the CMC@Zn electrode exhibited pronounced and disordered growth of Zn dendrites, along with fibrous structures attributed to detached SiO₂ from the glass fiber separator. This morphology is comparable to that of the severe Zn dendrites on cycled bare Zn (**Fig. 3a**), suggesting that CMC alone provides negligible benefit in regulating Zn deposition. In contrast, LAPO@Zn exhibits smooth, vertically aligned Zn deposits (**Fig. 3b**), indicating that the LAPONITE component plays a decisive role in promoting uniform Zn plating.

Second, linear sweep voltammetry (LSV) and linear polarization measurements were carried out to evaluate HER activity and corrosion behavior. Linear polarization was conducted at a scan rate of 10 mV s⁻¹ using Zn foil as the working electrode, Pt as the counter electrode, and Ag/AgCl as the reference electrode. LSV curves were obtained from Zn||Ti half cells at a scan rate of 5 mV s⁻¹ to assess the HER activity on substrates with different coatings. As shown in **Supplementary Fig. 33a**, the HER onset potentials for LAPO@Zn, CMC@Zn, and bare Zn are -0.145 V, -0.062 V, and -0.036 V, respectively. This demonstrates that while CMC slightly suppresses HER compared to bare Zn, its effect remains limited and is clearly inferior to that of LAPO@Zn. Moreover, the corrosion current density shown in **Supplementary Fig. 33b** for CMC@Zn (13.03 μA cm⁻²) is significantly higher than that of LAPO@Zn (1.54 μA cm⁻²), reinforcing the superior corrosion resistance provided by LAPONITE.

Third, electrochemical reversibility was evaluated in Zn||Cu cells. As shown in **Supplementary Fig. 34**, the CMC@Zn electrode exhibited stable cycling for less than 50 h—slightly improved over bare Zn, but inferior to LAPO@Zn, which maintained stable cycling for over 200 h (**Supplementary Fig. 21**).

Collectively, these results clearly demonstrate that CMC alone contributes only marginally to the suppression of HER and dendritic Zn growth. The beneficial effects observed in LAPO@Zn stem primarily from the functional role of LAPONITE, which offers Zn²⁺/SO₄²⁻ separation capability and ionic flux regulation. While CMC alone offers limited electrochemical benefit, its presence in the composite serves as an effective binder. The combination with LAPONITE helps enhance the interfacial

compatibility between the interlayer and the Zn metal surface and improves the mechanical integrity of the composite coating at the microscale.

We have included these new findings on **Page 37 of the revised Supplementary Information**, and the relevant discussion has been **marked in red**:

Supplementary Note 8. Role of CMC without LAPONITE.

To isolate the effect of CMC, we prepared Zn electrodes coated solely with CMC (CMC@Zn) at the same loading (1 mg cm^{-2}) to the LAPONITE layer of LAPO@Zn. SEM images after Zn plating revealed disordered dendrite formation and detached SiO_2 fibers (Supplementary Fig. 32), similar to those observed on bare Zn (Fig. 3a in the main text), indicating limited regulation of Zn deposition. Furthermore, CMC@Zn exhibited higher HER activity (with a more positive onset potential) and corrosion current density compared to LAPO@Zn (Supplementary Fig. 33). In Zn||Cu cells, CMC@Zn failed within 50 h (Supplementary Fig. 34), whereas LAPO@Zn maintained stable cycling for over 200 h (Supplementary Fig. 21). These results confirm that the uniform deposition and improved reversibility mainly originate from the LAPONITE component, rather than CMC.

Supplementary Figure 32. SEM images of the CMC@Zn foils after 1 h plating at 1 mA cm^{-2} .

Supplementary Figure 33. a, Linear sweep voltammetry (LSV) curves of bare Ti, CMC@Ti, and LAPO@Ti electrodes. **b**, Linear polarization curves of bare Zn, CMC@Zn, and LAPO@Zn electrodes.

Supplementary Figure 34. Coulombic efficiency (CE) profile of Zn||Cu coin cells using CMC@Cu electrodes at 2 mA cm^{-2} , with a fixed capacity of 1 mAh cm^{-2} .

Comment 2-2:

During continuous plating/stripping, do zinc ions deposit not only beneath the coating, but also in the middle and on top of the coating?

Response to Comment 2-2:

We sincerely thank Reviewer #2 for this insightful question. To examine whether Zn deposits from not only beneath but also within or atop the LAPONITE coating, we conducted XPS measurements on the top and middle regions of the LAPONITE+CMC interlayer after 20 cycles of Zn plating/stripping at 1 mA cm^{-2} for 1 h (**Supplementary Fig. 31**). The “top” region refers to the unmodified surface of the interlayer, while the “middle” region was obtained by mechanically scraping off the top half of the coating before measurement.

The measured Zn $2p_{3/2}$ binding energies were 1022.4 eV (top) and 1022.6 eV (middle), both significantly higher than the characteristic value for metallic Zn^0 (~ 1021.5 eV, referred to Adv. Energy Mater. 2022, 12, 2202784; Adv. Mater. 2024, 36, 2406257; ACS Appl. Mater. Interfaces 2023, 15, 26718) and thus more consistent with Zn^{2+} species. Given that LAPONITE ($\text{Na}_{0.2}(\text{Mg},\text{Li})_3\text{Si}_4\text{O}_{10}(\text{OH})_2 \cdot 4\text{H}_2\text{O}$) is a typical insulating nanosilicate with negligible electronic conductivity, Zn^{2+} ions are unlikely to be reduced within or on top of the coating. Instead, Zn metal deposition is expected to occur after Zn^{2+} ions migrate through the interlayer and reach the current collector (e.g., Zn or Cu foil), in accordance with common insulating coating design principles for metal anodes (referred to Adv. Mater. 2022, 34, 2202188; Adv. Energy Mater. 2024, 24, 2401018; Adv. Funct. Mater. 2021, 31, 2104361).

Furthermore, the slightly higher binding energy in the middle region (1022.6 eV) may indicate more extensive coordination between Zn^{2+} and the surrounding LAPONITE matrix within the film interior, as compared to the more loosely bound Zn^{2+} at the surface.

We have included these new findings in **Page 36 of the revised Supplementary Information**, and the relevant discussion has been **marked in red**.

Supplementary Note 7. Absence of Zn deposition within or atop the LAPONITE coating.

Supplementary Figure 31. XPS spectra of the LAPONITE+CMC interlayer after 20 deposition/stripping cycles at 1 mA cm⁻² for 1 h. The red and blue curves correspond to the top surface and middle region of the coating, respectively.

Both spectra show Zn²⁺ signals with no evidence of metallic Zn (Zn⁰, which typically exhibits a peak at around 1021.5 eV), indicating that Zn reduction occurs only after Zn²⁺ passes through the LAPONITE coating and reaches the conductive substrate.

Comment 2-3:

According to Fig. 1b, what drives the migration of zinc ions through the bending path between LAPONITE layers? Does the bending migration path affect the zinc ion migration, leading to deposition between the layers?

Response to Comment 2-3:

We sincerely thank Reviewer #2 for this thoughtful question. For the first part, the

migration of Zn^{2+} ions through the bending paths between LAPONITE layers is driven by the internal electric field during battery discharging, which directs Zn^{2+} ions from the electrolyte towards the Zn anode through the interlayer structure.

Regarding whether Zn deposition occurs within the LAPONITE layers, this is an important point. LAPONITE ($\text{Na}_{0.2}(\text{Mg},\text{Li})_3\text{Si}_4\text{O}_{10}(\text{OH})_2 \cdot 4\text{H}_2\text{O}$) is a typical nanosilicate with very low electronic conductivity. Due to this insulating nature, Zn^{2+} ions are not expected to be within or on the LAPONITE interlayer and therefore cannot be reduced to metallic Zn in this region. Instead, reduction to Zn^0 occurs only after Zn^{2+} ions pass through the coating and reach the conductive substrate (e.g., Zn or Cu foil). This behavior aligns with established design principles for insulating coatings used in Zn metal anodes (Adv. Mater. 2022, 34, 2202188; Adv. Energy Mater. 2024, 24, 2401018; Adv. Funct. Mater. 2021, 31, 2104361).

This conclusion is further supported by XPS analysis (**Supplementary Fig. 31**), which shows Zn $2p_{3/2}$ binding energies of 1022.4 eV at the top surface and 1022.6 eV in the interior of the LAPONITE+CMC interlayer after 20 plating/stripping cycles. These values are significantly higher than the characteristic value of metallic Zn (~1021.5 eV, referred to Adv. Energy Mater. 2022, 12, 2202784; Adv. Mater. 2024, 36, 2406257; ACS Appl. Mater. Interfaces 2023, 15, 26718) and thus correspond to Zn^{2+} species. The insulation nature of the interlayer and the absence of Zn^0 signals confirm that Zn deposition takes place only beneath the coating after Zn^{2+} reaches the conductive substrate.

We have included these new findings in **Page 36 of the revised Supplementary Information**, and the relevant discussion has been **marked in red**.

Supplementary Note 4. Absence of Zn deposition within or atop the LAPONITE coating.

Supplementary Figure 31. XPS spectra of the LAPONITE+CMC interlayer after 20 deposition/stripping cycles at 1 mA cm⁻² for 1 h. The red and blue curves correspond to the top surface and middle region of the coating, respectively.

Both spectra show Zn²⁺ signals with no evidence of metallic Zn (Zn⁰, which typically exhibits a peak at around 1021.5 eV), indicating that Zn reduction occurs only after Zn²⁺ passes through the LAPONITE coating and reaches the conductive substrate.

Comment 2-4:

How were the RSS data in Fig. 2d and e obtained?

Response to Comment 2-4:

We sincerely thank Reviewer #2 for pointing out this important detail, which helps us avoid any ambiguity in our data interpretation. The residual sum of squares (RSS) shown in Fig. 2d and 2e is a mathematical metric used to quantify the degree of

deviation of local Zn^{2+} flux from its in-plane average across the electrode surface. It does not carry direct physical meaning but serves as a numerical measure of flux uniformity.

If the in-plane flux were perfectly uniform, each normalized value of N/N_{avg} would be 1, and the deviation $(N/N_{\text{avg}}-1)$ would be zero at all positions. In our plots, the vertical axis represents the absolute value of this deviation. To compute the RSS, we squared each deviation value, i.e., $|N/N_{\text{avg}}-1|^2$, and summed the results across the entire surface. A larger RSS indicates greater non-uniformity in Zn^{2+} flux, whereas a smaller RSS reflects a more uniform flux distribution.

We have updated the figure caption of **Fig. 2d and 2e** accordingly to clearly describe this method. The revisions are **marked in red** in the revised manuscript.

Fig. 2 | Diffusion progress on different electrodes. d,e, Simulated Zn^{2+} flux density on bare Zn (d) and LAPO@Zn (e) electrode surfaces. N and N_{avg} represent the local and average Zn^{2+} flux densities, respectively. The y-axis shows the absolute deviation of normalized flux $(N/N_{\text{avg}}-1)$. RSS (residual sum of squares) was calculated by squaring each deviation and summing across the surface. Data collected from ionic concentration field simulations in Supplementary Fig. 7.

Comment 2-5:

The temperature-dependent EIS data in Fig. 2f is too limited, and the activation energy fitting is not precise enough.

Response to Comment 2-5:

We sincerely thank Reviewer #2 for raising this important concern, which helps us

eliminate potential ambiguity in our experimental description. In our initial temperature-dependent EIS tests, the coin cells were sealed in pouches and submerged in a water bath, with the temperature controlled via an external heating rod soaking within the water bath. However, this setup introduced certain limitations in maintaining stable and uniform heating, which may have affected the precision of the fitting results.

To improve data accuracy and ensure in-situ measurement conditions, we repeated the EIS tests using an AutoLab PGSTAT302N system, which enables direct and uniform heating of the coin cells with integrated temperature control. The frequency range used was 0.01-10⁵ Hz. As shown in the updated EIS spectra (new **Supplementary Fig. 11**), we obtained improved impedance profiles for both bare Zn and LAPO@Zn at various temperatures. Based on these new datasets, we reconstructed the Arrhenius plots (new **Fig. 2f**), which yielded activation energies of 46.89 kJ mol⁻¹ for bare Zn and 33.32 kJ mol⁻¹ for LAPO@Zn.

These updated results are consistent with our original conclusion that the LAPO@Zn interface exhibits enhanced charge transfer kinetics. We have revised the main figure and added new temperature-dependent EIS data to the revised Supplementary Information. The relevant revisions are **marked in red** in the revised manuscript:

Page 16, Line 21 of the revised manuscript:

Electrochemical measurements

Temperature-dependent EIS measurements were conducted on a AutoLab PGSTAT302N system.”

Fig. 2 | f, Arrhenius plots of R_{ct}^{-1} values at different temperatures (from 20 °C to 70 °C). The activation energies for the interfacial charge transfer processes were determined by calculating the slope of the fitted line.

Supplementary Figure 11. EIS spectra at different temperatures. a, Bare Zn symmetric cells. **b**, LAPO@Zn symmetric cells.

Comment 2-6:

Line 195, the significance of the physical and chemical parameters such as a, b, c3DI, c3DP, d3DI, d3DP, I0, τ_{DL} is unclear.

Response to Comment 2-6:

We sincerely thank Reviewer #2 for raising this important point. To clarify the physical

significance of the parameters a , b , c_{3DI} , c_{3DP} , d_{3DI} , d_{3DP} , I_0 , and τ_{DL} , we have provided their complete analytical expressions and physicochemical definitions in **Supplementary Note 3** of the revised Supplementary Information (marked in red).

Notably, c_{3DI} and c_{3DP} share the same analytical expression ($c = nFAk_3$), and d_{3DI} and d_{3DP} also follow an identical form ($d = \pi M^2 N_0 k_1^2 / \rho^2$). The key difference between I_{3DI} and I_{3DP} lies in the time exponent (t^2 vs. t^3), which corresponds to different nucleation mechanisms (instantaneous vs. progressive). Therefore, for clarity and conciseness, we have used the unified notations c and d in the main text. These relationships are now clearly presented in both the revised main text and revised Supplementary Information:

Page 8, Line 8 of the revised manuscript:

$$I_{DL} = I_0 \cdot e^{\frac{t}{\tau_{DL}}} \quad (1)$$

$$I_{2DI} = a \cdot t \cdot e^{-b \cdot t^2} \quad (2)$$

$$I_{3DI} = c \cdot (1 - e^{-d \cdot t^2}) \quad (3)$$

$$I_{3DP} = c \cdot (1 - e^{-d \cdot t^3}) \quad (4)$$

The parameters a , b , c , d , I_0 , and τ_{DL} are related to physicochemical quantities²⁹. Detailed expressions are provided in Supplementary Note 3.

Page 31 of the revised Supplementary Information:

Supplementary Note 3. physical meaning of the parameters used in CA analysis.

The parameters used in the CA analysis are defined as follows:

$$a = \frac{2 \cdot \pi \cdot n \cdot F \cdot M \cdot h \cdot A \cdot N_0 \cdot k_1^2}{\rho} \quad (3)$$

$$b = \frac{\pi \cdot M^2 \cdot N_0 \cdot k_1^2}{\rho^2} \quad (4)$$

$$c = n \cdot F \cdot A \cdot k_3 \quad (5)$$

$$d = \frac{\pi \cdot M^2 \cdot N_0 \cdot k_2^2}{\rho^2} \quad (6)$$

$$I_0 = \frac{\Delta E}{R} \quad (7)$$

$$\tau_{DL} = R \cdot C \quad (8)$$

Definition of symbols:

N_0 — number density of 2D or 3D active sites formed instantaneously;

k_1 — rate constant of the 2D growth of the nucleus;

k_2 — rate constant of the 3D parallel (lateral) growth;

k_3 — rate constant of the 3D perpendicular (outward) growth;

A — electrode surface area active for 2D or 3D nucleation and growth;

h — height of the 2D layer;

ρ — density of the deposited 2D layer or 3D layer;

M — molar mass;

n — number of electrons transferred in the deposition reaction;

ΔE — direct current potential step applied to the electrode;

R — solution resistance (equivalent series resistance);

C — double layer capacitance (equivalent series capacitance);

Comment 2-7:

Please provide the XRD data of the original zinc foil, the gradient process of the 100 crystal plane dominant orientation after LAPO@Zn cycling, and the XRD data for the 002, 100, 101, and other crystal planes.

Response to Comment 2-7:

We sincerely thank Reviewer #2 for this valuable suggestion. To address this point, we performed additional GIXRD measurements on the original zinc foil and LAPO@Zn electrodes at various deposition and cycling stages. The results are presented in

Supplementary Fig. 28, which includes the diffraction regions of Zn (002), (100), (101), (102), (103), and (110) crystal planes.

As recommended, we first provide the GIXRD pattern of the original zinc foil (i.e., the “Pristine” curve in **Supplementary Fig. 28**), showing an initial intensity ratio of $I_{(002)}/I_{(100)} = 3.05$.

Upon deposition at 1 mA cm^{-2} for 10, 20, 30, and 60 minutes, the $I_{(002)}/I_{(100)}$ ratios evolve to 2.69, 1.62, 1.59, and 0.54, respectively. This steady decrease reflects a preferential enhancement of the (100) plane during Zn growth. As discussed in the main text, this reorientation is attributed to the LAPONITE coating, which directs Zn^{2+} flux through its lamellar channels and charge-separated sites, promoting controlled and uniform deposition with increasing (100) texture.

After one full deposition/stripping cycle (1 h plating followed by 1 h stripping), the $I_{(002)}/I_{(100)}$ ratio slightly increases to 0.67, likely due to the preferential dissolution of Zn grains with dominant (100) planes during the stripping process.

After 50 and 100 cycles, the $I_{(002)}/I_{(100)}$ ratios gradually increase to 0.91 and 1.01, respectively. This slightly increase compared to the value of 0.67 after one cycle may arise from the progressive weakening of LAPONITE’s orientation-guiding effect due to minor interfacial degradation and accumulation of byproducts such as zinc hydroxide sulfate (ZHS) on the Zn surface. In addition, the inevitable hydrogen evolution reaction (HER) under mildly acidic conditions may disrupt the local interfacial structure. These side effects are thermodynamically feasible in dilute aqueous electrolytes and have been widely reported (such as in Chem. Soc. Rev. 2020, 49, 4203).

We have added the corresponding discussion to the **revised Supplementary Information** and **marked the new content in red**.

Supplementary Note 5. GIXRD analysis of Zn orientation evolution during deposition and cycling on LAPO@Zn.

Supplementary Figure 28. GIXRD patterns of LAPO@Zn at different stages, including pristine condition (also the same to the original zinc foil), after 10 min, 20 min, 30 min, and 60 min deposition at 1 mA cm^{-2} , as well as after 1, 50, and 100 cycles of deposition/stripping cycles, respectively.

The $I_{(002)}/I_{(100)}$ ratio decreases from 3.05 (pristine) to 2.69, 1.62, 1.59, and 0.54 after 10, 20, 30, and 60 min of deposition at 1 mA cm^{-2} , indicating a gradual enhancement of the (100) orientation guided by the LAPONITE coating. After 1, 50, and 100 cycles, the ratio gradually increases to 0.67, 0.91, and 1.01, respectively, likely due to weakened orientation control caused by interfacial degradation from HER and the accumulation of ZHS byproducts.

Comment 2-8:

The SEM images in Fig. 3b are not clear enough, making it difficult to identify the dominant orientation of the 100 crystal plane.

Response to Comment 2-8:

We sincerely thank Reviewer #2 for this valuable suggestion. We fully agree that the SEM image in the original Fig. 3b was not sufficiently clear to directly illustrate the preferential (100) orientation induced by the LAPONITE coating. This is likely because the original image was taken with the LAPONITE+CMC interlayer still present on the Zn surface, which obscured the deposited Zn and made the underlying morphology less distinguishable.

To address this, we removed the surface LAPONITE+CMC layer and re-examined the exposed Zn morphology after 1 h of plating at 1 mA cm^{-2} . As shown in **Supplementary Fig. 27**, the underlying Zn structure clearly exhibits vertically aligned, compact columnar grains—supporting the conclusion that the LAPONITE coating promotes Zn deposition along the (100) plane.

This morphology is consistent with previously reported vertically stacked Zn growth (Nat. Commun. 2022, 13, 3252), although achieved via different approaches. Our results further confirm that such vertical alignment is beneficial for stable and reversible Zn plating/stripping. Please find the new **Supplementary Note 4** marked in red in the revised **Supplementary Information** for further clarification.

Supplementary Note 4. Zn morphology beneath the LAPONITE coating.

Supplementary Figure 33. SEM images of the LAPO@Zn electrodes after 1 h plating at 1 mA cm^{-2} . The images were collected after removing the surface LAPONITE layer.

The exposed Zn surface displays compact, vertically aligned grains, directly confirming

the guided growth along the (100) plane. This orientation is favorable for suppressing dendrites and enhancing cycling reversibility.

Comment 2-9:

The description of deposition orientation in Fig. 3d and e is not clear enough.

Response to Comment 2-9:

We sincerely thank Reviewer #2 for pointing out this issue. We have revised the description of the deposition orientation in Fig. 3d and 3e for clarity. The revised content is marked in red in the revised manuscript.

Page 9, Line 5 of the revised manuscript:

In-situ X-ray computed tomography (X-CT) inspection provides further insight into the deposition orientation. After substantial Zn deposition on bare Zn and LAPO@Zn substrates, we collected their corresponding X-CT images and performed segmentation of all Zn deposits above the original substrates (Supplementary Fig. 14). Over 3,000 units were identified from each electrode and mapped onto a polar axial system (Fig. 3d,e). The orientation of each Zn deposit was characterized by two angles: the in-plane angle (ranging from 0 to 360 degrees) and the out-of-plane angle (ranging from 0 to 90 degrees; angles beyond 180 degrees are physically irrelevant due to the upward growth direction of Zn, and 135 degrees is equivalent to 45 degrees owing to axial symmetry). We primarily focus on the out-of-plane angle to determine whether the Zn deposits are vertically, horizontally, or obliquely stacked. As shown in Fig. 3d, the out-of-plane direction angle distribution of Zn deposits on bare Zn is nearly uniform across all radial directions, indicating randomly oriented growth and a lack of spatial control. In contrast, Fig. 3e reveals that Zn deposits on LAPO@Zn are predominantly oriented near 0 and 90 degrees, forming a bimodal distribution corresponding to the smallest and largest circles in the polar plot. This reflects a more ordered and vertically aligned deposition pattern. The non-destructive X-CT analysis offers critical insight into how

LAPONITE's layered channels and charge-separated structure guide Zn^{2+} , favoring growth along the (100) plane, which typically aligns at an angle of 70-90 degrees relative to the substrate (i.e., more vertically stacked).

Comment 2-10:

Mineral materials with distinct positive and negative charge centers are not limited to LAPONITE. Do other clay materials such as attapulgite, kaolinite, and montmorillonite exhibit similar effects?

Response to Comment 2-10:

We sincerely thank Reviewer #2 for raising this insightful question. As correctly noted, mineral materials with intrinsic charge anisotropy are not limited to LAPONITE. To investigate whether other naturally occurring clays can induce similar effects, we conducted a series of control experiments using attapulgite, kaolinite, and montmorillonite as coating layers on Zn anodes, following the same fabrication protocol.

As shown in the SEM images and Zn||Zn cycling results (**Figure R1**), all three clay coatings improve the Zn deposition morphology compared to bare Zn (Fig. 3a in the main text), effectively suppressing the formation of sharp dendrites. They also enhance cycling stability under identical galvanostatic conditions (**Figure R2**). For instance, Zn||Zn symmetric cells using attapulgite, kaolinite, and montmorillonite coatings stably cycled for 200 h, whereas the bare Zn cell failed after 94 h (Fig. 5a in the main text).

However, these three clays still perform less effectively than LAPONITE. After around 200 cycles, the symmetric cells coated with attapulgite, kaolinite, and montmorillonite show polarization voltages of 42.5 mV, 34.7 mV, and 36.6 mV, respectively, which remain higher than that of the LAPONITE-coated Zn (19.4 mV, as shown in Fig. 5a). This difference may be attributed to LAPONITE's unique structure with nanoscale thickness, high surface charge density, and highly ordered layer

alignment. These features help achieve denser interfacial coverage, more uniform Zn^{2+} flux, and reduced reaction polarization during cycling.

These comparisons confirm that other charged clays may contribute to improved Zn deposition and further highlight the exceptional effectiveness of LAPONITE in achieving highly stable and uniform growth. We sincerely thank the reviewer for the thoughtful suggestion.

Figure R1. SEM images of the Zn electrodes coated with attapulgite, kaolinite, and montmorillonite, after 1 h plating at 1 mA cm^{-2} . The images were collected after removing the surface layers.

Figure R2. Galvanostatic cycling curves of Zn||Zn symmetric cells at 1 mA cm^{-2} with a fixed capacity of 1 mAh cm^{-2} . The Zn anodes are coated with attapulgite (a), kaolinite (b), and montmorillonite (c).

Comment 2-11:

The microscopic morphology and macroscopic features (such as thickness) of the coating are not clear enough. Additionally, does the coating thickness and proportion have an impact?

Response to Comment 2-11:

We sincerely thank Reviewer #2 for this constructive comment. To better elucidate the microscopic and macroscopic features of the LAPONITE+CMC interlayer and their impact on electrochemical performance, we systematically varied the coating thickness by adjusting the LAPONITE mass loading to 0.5, 1, and 2 mg cm⁻², while keeping the LAPONITE/CMC ratio fixed at 1:1 (wt%), consistent with the protocol described in the main text. Since this interlayer was used on both Zn and Cu substrates for cycling and Coulombic efficiency tests, we characterized its structure by peeling off the coatings (as illustrated in Fig. 1c of the main text) and transferring the freestanding films onto conductive tape for SEM cross-sectional imaging.

As shown in **Supplementary Fig. 38**, the 0.5 mg cm⁻² coating exhibits a relatively compact and uniform center region with a thickness of 6.67 μm (**Supplementary Fig. 38a**), but the edge region is significantly thicker and more irregular, measuring 8.68 μm (**Supplementary Fig. 38b**), which corresponds to a 30.13% increase. In addition, small cracks and film curling are observed at the edge, indicating poor uniformity in both the lateral and vertical directions. This could be attributed to insufficient CMC content or the overall low mass loading, which can compromise the mechanical integrity of the coating during coating layer formation.

In contrast, the 2 mg cm⁻² coating exhibits pronounced wrinkling and cracking across the entire film (**Supplementary Fig. 38e,f**). The edge region becomes substantially thicker than the center (18.62 μm vs. 12.88 μm), resulting in a 44.57% difference. This uneven thickness is likely caused by gravitational effects and lateral diffusion during drying. These structural inconsistencies indicate that excessive mass loading disrupts film continuity and compromises overall uniformity.

The 1 mg cm⁻² coating—the composition used throughout the original main manuscript—displays the most homogeneous morphology, with center and edge thickness of 9.22 μm and 10.02 μm, respectively, corresponding to only an 8.78% variation (**Supplementary Fig. 38c,d**). Moreover, well-distributed vertical gaps are observed throughout the film, which may facilitate effective cation-anion separation by the LAPONITE lamellae.

For the cycling performance, the 1 mg cm⁻² interlayer enables the most stable Zn||Zn cycling, sustaining over 1,200 h of operation (Fig. 5a in the main text). In contrast, the 0.5 and 2 mg cm⁻² interlayers fail before 500 h (**Supplementary Fig. 39**). These findings suggest that the 1 mg cm⁻² configuration offers the most balanced combination of mechanical integrity, structural uniformity, and electrochemical performance.

This additional analysis has deepened our understanding of the key structural characteristics required for an effective LAPONITE+CMC interlayer. We thank the reviewer again for highlighting this important aspect. The corresponding data and discussion have been added to the **revised Supplementary Information** and are **marked in red**:

Page 42 of the revised Supplementary Information:

Supplementary Note 10. Effect of thickness of the LAPONITE+CMC coating.

Supplementary Figure 38. Cross-sectional SEM images of LAPONITE+CMC

coatings with different LAPONITE mass loading of (a,b) 0.5 mg cm^{-2} , (c,d) 1 mg cm^{-2} , and (e,f) 2 mg cm^{-2} . SEM images collected from the center region are shown in a,c,e, and those from the edge region are shown in b,d,f.

Supplementary Figure 39. Galvanostatic cycling curves of LAPO@Zn||LAPO@Zn symmetric cells at 1 mA cm^{-2} with a fixed capacity of 1 mAh cm^{-2} . The mass loadings of the LAPONITE are 0.5 mg cm^{-2} and 2 mg cm^{-2} , respectively. Each sample was tested twice to confirm reproducibility.

Cross-sectional SEM image (Supplementary Fig. 38) reveals that the 1 mg cm^{-2} LAPONITE+CMC interlayer exhibits superior structural uniformity, with a center thickness of $9.22 \text{ }\mu\text{m}$ and an edge thickness of $10.02 \text{ }\mu\text{m}$ (8.78% variation). Vertically aligned gaps are observed across the coating layer, which may facilitate directional ion transport and effective $\text{Zn}^{2+}/\text{SO}_4^{2-}$ separation. In contrast, the 0.5 mg cm^{-2} coating displays a 30.13% thickness variation between center ($6.67 \text{ }\mu\text{m}$) and edge ($8.68 \text{ }\mu\text{m}$), along with edge curling and microcracks. The 2 mg cm^{-2} coating shows greater variation of 44.57% ($12.88 \text{ }\mu\text{m}$ at the center vs. $18.62 \text{ }\mu\text{m}$ at the edge), and widespread wrinkling and cracking. These structural inconsistencies are indicative of poor coating layer formation and compromised mechanical robustness. Correspondingly, the 1 mg cm^{-2}

interlayer enables the most stable cycling performance, maintaining over 1,200 h in Zn||Zn symmetric cells (Fig. 5a in the main text), whereas the other two configurations fail before 500 h (Supplementary Fig. 39).

Comment 2-12:

There are formatting errors in the citation of ref2 and ref20 in the introduction.

Response to Comment 2-12:

We sincerely thank Reviewer #2 for kindly pointing out the formatting issues. The citation errors associated with Ref. 2 and Ref. 20 (which now appears as Ref. 21 in the revised manuscript due to reference reordering) in the Introduction have been corrected. These changes have been clearly **marked in red** in the revised manuscript:

Page 2, Line 13 of the revised manuscript:

For example, the United States aims for 50% of vehicle sales to be electric by 2030, while the European Union targets a 55% reduction in greenhouse gas emissions by 2030 compared to 1990 levels^{2,3}.

Page 3, Line 10 of the revised manuscript:

In ZnSO₄ aqueous electrolyte, for instance, Zn²⁺-SO₄²⁻ contact ion pairs (CIPs) form readily owing to the divalent chemistry of Zn²⁺ ions²¹.

Comment 2-13:

Supplementary Fig. 24-27 are not described in the main text.

Response to Comment 2-13:

We sincerely thank Reviewer #2 for bringing this to our attention. In response, we have thoroughly revised the manuscript to ensure that all Supplementary Figures (including

original Supplementary Figs. 24-27) and other newly added Supplementary Figures and Supplementary Notes, are now properly cited and discussed in the revised manuscript.

Response to Reviewer #3:

General Comments:

This manuscript presents significant contributions to Zn anode stabilization and dendrite suppression. The scientific rationale is sound, and the experimental techniques are advanced. However, major improvements are required in the clarity of writing, data interpretation, and additional experimental validation of theoretical claims. Addressing these points will strengthen the manuscript's impact and relevance.

Response:

We sincerely thank Reviewer #3 for recognizing the significance of our contributions to Zn anode stabilization and dendrite suppression. We appreciate the Reviewer's constructive feedback regarding the clarity of writing, data interpretation, and the need for further experimental validation. In response, we have thoroughly revised the manuscript to improve its presentation and have incorporated additional data and explanations to support the theoretical claims. We hope that these revisions have addressed the Reviewer's concerns and further strengthened the clarity and impact of the manuscript.

Comment 3-1:

The anode surface modification was done by a mixture of LAPONITE and CMC, but the manuscript only discusses LAPONITE. It is well-known that CMC is also efficient in preventing dendrite formation and parasitic side reactions. The surface layer thickness is not provided.

Response to Comment 3-1:

We sincerely thank Reviewer #3 for raising this important question, which has greatly helped us improve the rigor and scientific clarity of our manuscript. To investigate the influence of CMC alone on Zn deposition uniformity and suppression of water-induced

side reactions, we conducted a series of control experiments by coating Zn foil with CMC alone (denoted as CMC@Zn), using the same areal loading (1 mg cm^{-2}) as the LAPONITE-CMC composite layer applied in LAPO@Zn.

First, the Zn deposition morphology was examined via SEM after plating 1 mAh cm^{-2} of Zn at 1 mA cm^{-2} (**Supplementary Fig. 32**). As shown, the CMC@Zn electrode exhibited pronounced and disordered growth of Zn dendrites, along with fibrous structures attributed to detached SiO_2 from the glass fiber separator. This morphology is comparable to that of the severe Zn dendrites on cycled bare Zn (**Fig. 3a**), suggesting that CMC alone provides negligible benefit in regulating Zn deposition. In contrast, LAPO@Zn exhibits smooth, vertically aligned Zn deposits (**Fig. 3b**), indicating that the LAPONITE component plays a decisive role in promoting uniform Zn plating.

Second, linear sweep voltammetry (LSV) and linear polarization measurements were carried out to evaluate HER activity and corrosion behavior. Linear polarization was conducted at a scan rate of 10 mV s^{-1} using Zn foil as the working electrode, Pt as the counter electrode, and Ag/AgCl as the reference electrode. LSV curves were obtained from Zn||Ti half cells at a scan rate of 5 mV s^{-1} to assess the HER activity on substrates with different coatings. As shown in **Supplementary Fig. 33a**, the HER onset potentials for LAPO@Zn, CMC@Zn, and bare Zn are -0.145 V , -0.062 V , and -0.036 V , respectively. This demonstrates that while CMC slightly suppresses HER compared to bare Zn, its effect remains limited and is clearly inferior to that of LAPO@Zn. Moreover, the corrosion current density shown in **Supplementary Fig. 33b** for CMC@Zn ($13.03 \text{ } \mu\text{A cm}^{-2}$) is significantly higher than that of LAPO@Zn ($1.54 \text{ } \mu\text{A cm}^{-2}$), reinforcing the superior corrosion resistance provided by LAPONITE.

Third, electrochemical reversibility was evaluated in Zn||Cu cells. As shown in **Supplementary Fig. 34**, the CMC@Zn electrode exhibited stable cycling for less than 50 h—slightly improved over bare Zn, but inferior to LAPO@Zn, which maintained stable cycling for over 200 h (**Supplementary Fig. 21**).

Collectively, these results clearly demonstrate that CMC alone contributes only marginally to the suppression of HER and dendritic Zn growth. The beneficial effects observed in LAPO@Zn stem primarily from the functional role of LAPONITE, which

offers $\text{Zn}^{2+}/\text{SO}_4^{2-}$ separation capability and ionic flux regulation. While CMC alone offers limited electrochemical benefit, its presence in the composite serves as an effective binder. The combination with LAPONITE helps enhance the interfacial compatibility between the interlayer and the Zn metal surface and improves the mechanical integrity of the composite coating at the microscale.

We have included these new findings on **Page 37 of the revised Supplementary Information**, and the relevant discussion has been **marked in red**:

Supplementary Note 8. Role of CMC without LAPONITE.

To isolate the effect of CMC, we prepared Zn electrodes coated solely with CMC (CMC@Zn) at the same loading (1 mg cm^{-2}) to the LAPONITE layer of LAPO@Zn. SEM images after Zn plating revealed disordered dendrite formation and detached SiO_2 fibers (Supplementary Fig. 32), similar to those observed on bare Zn (Fig. 3a in the main text), indicating limited regulation of Zn deposition. Furthermore, CMC@Zn exhibited higher HER activity (with a more positive onset potential) and corrosion current density compared to LAPO@Zn (Supplementary Fig. 33). In Zn||Cu cells, CMC@Zn failed within 50 h (Supplementary Fig. 34), whereas LAPO@Zn maintained stable cycling for over 200 h (Supplementary Fig. 21). These results confirm that the uniform deposition and improved reversibility mainly originate from the LAPONITE component, rather than CMC.

Supplementary Figure 32. SEM images of the CMC@Zn foils after 1 h plating at 1 mA cm^{-2} .

Supplementary Figure 33. a, Linear sweep voltammetry (LSV) curves of bare Ti, CMC@Ti, and LAPO@Ti electrodes. **b**, Linear polarization curves of bare Zn, CMC@Zn, and LAPO@Zn electrodes.

Supplementary Figure 34. Coulombic efficiency (CE) profile of Zn||Cu coin cells using CMC@Cu electrodes at 2 mA cm^{-2} , with a fixed capacity of 1 mAh cm^{-2} .

We also appreciate Reviewer #3 for pointing out the omission of the surface layer thickness. We have now included this information in **Supplementary Fig. 38**. Cross-sectional SEM analysis of the 1 mg cm^{-2} LAPONITE+CMC interlayer shows a thickness of $9.22 \text{ }\mu\text{m}$ at the center and $10.02 \text{ }\mu\text{m}$ at the edge, corresponding to an in-plane variation of 8.78%. Furthermore, uniformly distributed and vertically aligned gaps are observed throughout the coating layer, which may promote directional Zn^{2+} transport and efficient $\text{Zn}^{2+}/\text{SO}_4^{2-}$ separation. These structural features contribute to the stable and long-term cycling performance of LAPO@Zn presented in the main text. Please see the updated figures **marked in red** on Page 36 of the revised Supplementary Information.

Supplementary Figure 38. Cross-sectional SEM images of LAPONITE+CMC coatings with different LAPONITE mass loading of (c,d) 1 mg cm^{-2} . c is the center region while d is the edge region.

Comment 3-2:

The manuscript presents compelling experimental data but lacks clear, concise transitions between sections. The introduction should better structure the challenges of Zn anodes and why ion flux vortices specifically affect deposition morphology.

Response to Comment 3-2:

We sincerely thank Reviewer #3 for the valuable suggestion and greatly appreciate the positive assessment of our experimental data as compelling. In response, we have revised the manuscript to improve the logical flow and transitions between major sections, ensuring that each dataset and analysis is better contextualized within the overall narrative. The specific revisions have been **marked in red** in the **updated manuscript** as outlined below:

1. End of Structural feature of LAPONITE section (Page 5, Line 15):

LAPONITE effectively reduces these $\text{Zn}^{2+}\text{-SO}_4^{2-}$ CIPs in the diffusion pathway, mitigating ion flux vortices and promoting a more uniform Zn^{2+} flux and deposition (Fig. 2c). **This mechanistic insight into CIP regulation and vortex suppression motivates a quantitative exploration of interfacial ion behavior via computational fluid dynamics (CFD) simulations.**

2. End of Joint CFD-electrochemical analysis section (Page 8, Line 21):

This rapid growth highlights the effectiveness of the LAPONITE coating in mitigating ion flux vortices, resulting in more uniform nucleation and controlled deposition. Consequently, LAPO@Zn reduces the tendency for dendrite formation and growth. **Although the simulations confirm improved Zn^{2+} flux uniformity, direct structural characterization is necessary to validate the morphological consequences of vortex suppression.**

3. End of Zn deposition orientation section (Page 9, Line 33):

This enhanced control reduces dendrite formation, especially the dendrite volume, branch size and curvature, contributing to improved anode uniformity and safety, which are essential for high-performance metal batteries. Beyond growth orientation, the compactness and grain structure of Zn deposits also play a comparably vital role in dendrite suppression and long-term stability.

4. End of Zn deposition compactness section (Page 12, Line 20):

These findings indicate that LAPONITE alleviates the increase in R_{ct} associated with ZHS accumulation, leading to a more stable interface compared to bare Zn. To validate the benefits of compact and uniform Zn deposits, electrochemical performance was evaluated in both symmetric cells and full cells.

In addition, for better structure the challenges of Zn anodes and why ion flux vortices specifically affect deposition morphology, we updated the introduction part as follows, the revision parts are marked in red:

Within the Introduction section (Page 2, Line 25):

However, the practical implementation of AZMBs is still hindered by significant challenges related to non-uniform Zn deposition and dendrite formation, which compromise cycle life and battery safety¹³⁻¹⁵. A key origin of these failures lies in chaotic ionic flows at the electrode interface¹⁶, which give rise to localized ion flux vortices. These vortices disrupt the uniformity of Zn nucleation and induce Zn^{2+} accumulation at pre-existing deposits, thereby facilitating spatially selective growth driven by local concentration gradients and electric field effects. Vortex-like ion transport has been reported in flow-driven systems, where external instabilities cause circulating flux and uneven deposition¹⁷. In static electrolytes, asymmetric ion migration may similarly induce localized rotational flux—hereafter referred to as an ion flux vortex—that disrupts interfacial deposition. The outcome is progressively uneven deposition and the emergence of angular, branched dendrites. This

morphological instability increases the likelihood of dendrites penetrating the separator, thus raising the risk of short-circuit failure. Addressing these issues requires both rational electrode material design and a deeper understanding of mass transport phenomena at the electrode-electrolyte interface¹⁸.

Comment 3-3:

Ion flux vortex: This term is introduced without sufficient context. Has it been previously reported in electrochemical literature, or is this a new concept introduced by the authors? Clarification is needed.

Response to Comment 3-3:

We appreciate Reviewer #3 for this insightful comment. The term Ion flux vortex builds upon previously reported observations of vortex-like ion motion near electrodes, particularly under electroconvective conditions. For example, Ma et al. (Sci. Adv. 2021, 7, eabf6941) visualized such flows in externally driven systems exhibiting overlimiting current and space-charge layer formation—also using Zn electrodes.

In contrast, the phenomenon described in our work arises under static aqueous electrolyte conditions. Here, we identified localized rotational ionic motion resulting from the asymmetric migration of contact ion pairs (Zn^{2+} - SO_4^{2-}) under an applied electric field. We define this as an ion flux vortex, which contributes to spatially non-uniform Zn deposition.

To clarify this concept, we have now added the following sentence to **the revised Introduction section marked in red**:

Page 2, Line 29 of the revised manuscript:

Vortex-like ion transport has been reported in flow-driven systems, where external instabilities cause circulating flux and uneven deposition¹⁷. In static electrolytes, asymmetric ion migration may similarly induce localized rotational flux—hereafter referred to as an ion flux vortex—that disrupts interfacial deposition. The outcome is

progressively uneven deposition and the emergence of angular, branched dendrites. This morphological instability increases the likelihood of dendrites penetrating the separator, thus raising the risk of short-circuit failure.

Comment 3-4:

(This translates into a 3.54-Ah Zn-MnO₂ pouch cell with stable performance over 100 cycles.) While promising, 100 cycles is relatively short for practical applications. Stable performance needs quantification—what capacity retention, Coulombic efficiency, or failure mode was observed?

Response to Comment 3-4:

We sincerely thank Reviewer #3 for raising this importance point. We fully agree that while 100 cycles is a meaningful milestone, it is still relatively short for practical applications. In response, we provide the following clarifications and enhancements regarding three key aspects: capacity retention, failure analysis, and capacity correction.

1. Capacity retention and Coulombic efficiency

As shown in the revised **Fig. 5c**, the LAPO@Zn||MnO₂ pouch cell shows an initial activation period, with the discharge capacity increasing from 2.75 Ah (6th cycle) to a peak of 3.17 Ah (44th cycle). By the 100th cycle, it retained 2.77 Ah, corresponding to 87.38% capacity retention relative to the maximum value. The Coulombic efficiency was 99.82% at the 100th cycle, with an average of 99.53% over 100 cycles. These metrics demonstrate excellent reversibility and cycling stability under practical full-cell conditions.

2. Post-mortem failure analysis

To investigate failure modes, we performed SEM, EDS, and XRD characterizations of both the anodes and cathodes extracted from Zn||MnO₂ pouch cells after 100 cycles (**Supplementary Figs. 35-37**). The bare Zn anode exhibited extensive randomly stacked dendrites (**Supplementary Fig. 35a-c**), along with pronounced corrosion, indicated by elevated oxygen (34.2%) and sulfur (6.1%) contents, which are likely

associated with the formation of $\text{Zn}_4\text{SO}_4(\text{OH})_6 \cdot n\text{H}_2\text{O}$ byproducts. In addition, a detectable silicon signal (2.3%) was observed, which is attributed to the detached glass fiber separator (**Supplementary Fig. 35d**). These results collectively suggest that the bare Zn anode underwent severe mechanical and chemical degradation, whereas such degradation was substantially mitigated in the LAPO@Zn anode.

In contrast, the LAPO@Zn anode retained vertically stacked Zn clusters (**Supplementary Fig. 36a**) with visible remnants of the LAPONITE coating (**Supplementary Fig. 36b**). Corrosion was significantly mitigated, as reflected in lower oxygen (28.9%) and sulfur (3.1%) contents in the EDX spectrum (**Supplementary Fig. 36d**). The presence of a magnesium signal in **Supplementary Fig. 36c** and a Mg content of 1.6% in the **Supplementary Fig. 36d** further confirm the retention of the LAPONITE layer on the Zn surface.

Regarding the MnO_2 cathodes, both configurations exhibited surface byproducts of $\text{Zn}_4\text{SO}_4(\text{OH})_6 \cdot 4\text{H}_2\text{O}$, resulting from irreversible H^+ intercalation into the MnO_2 cathodes (refer to Adv. Mater. 2022, 34, 2109092). Notably, the MnO_2 from the LAPO@Zn-based pouch cell showed only finer flake-like deposits (which is $\text{Zn}_4\text{SO}_4(\text{OH})_6 \cdot 4\text{H}_2\text{O}$, as shown in **Supplementary Fig. 37b,c**), while the bare Zn-coupled MnO_2 displayed larger, sheet-like structures (**Supplementary Fig. 37d,e**). These larger sheets correspond to the formation of $\text{Zn}_x\text{MnO}(\text{OH})_2$, which is indicative of Mn dissolution. The coexistence of $\text{Zn}_4\text{SO}_4(\text{OH})_6 \cdot 4\text{H}_2\text{O}$ and $\text{Zn}_x\text{MnO}(\text{OH})_2$ in the bare Zn-coupled MnO_2 (**Supplementary Fig. 37a**) originates from more severe anodic side reactions and greater pH fluctuations of the whole electrolyte, accelerating Mn dissolution at the cathode side (Nat. Commun. 2022, 13, 2371).

Overall, the dominant degradation mechanism in the LAPO@Zn system appears to be cathode-driven rather than anode-drive. To further extend cycle life beyond 100 cycles and approach practical application standards, we plan to develop modified, high-loading MnO_2 cathodes for use in the LAPO@Zn|| MnO_2 configuration in the future.

3. Correction of pouch cell capacity

We are especially grateful for this comment, which prompted us to carefully revisit our capacity calculation of the LAPO@Zn|| MnO_2 pouch cell. In our original manuscript,

we inadvertently reported a cathode mass loading of 5 mg cm^{-2} based on the initially intended experimental design. Upon reviewing our experimental records and equipment logs, we found that the actual cathode loading used in this manuscript was 4.47 mg cm^{-2} per side, corresponding to 8.94 mg cm^{-2} per double-coated MnO_2 cathode. For the 10-layer configuration in the assembled pouch cell, the total MnO_2 mass was 17,442.321 mg (confirmed by the original channel record as shown in **Figure R3**). We have also included the original cycling data for the LAPO@Zn|| MnO_2 (**Figure R4**), which clearly shows the full evolution of the whole discharge capacity (in Ah) during cycling. The maximum capacity was accurately measured as 3.17 Ah (44th cycle), rather than 3.54 Ah as initially reported.

We sincerely appreciate Reviewer #3's attention to detail, which helped us ensure the accuracy and transparency of our data. All corresponding figures and statements in the main text and Supplementary Information have been revised accordingly and are clearly **marked in red**. The list of updated contents are as follows:

Abstract section (Page 2, Line 6 of the revised manuscript):

This translates into a **3.17-Ah** Zn- MnO_2 pouch cell with stable performance over 100 cycles, offering a viable path toward scalable, high-performance metal-anode batteries.

End of Results section (Page 14, Line 3 of the revised manuscript):

As shown in Fig. 5c, the LAPO@Zn|| MnO_2 pouch cell, featuring the stabilized LAPO@Zn anode, delivered a **maximum** capacity of **3.17 Ah** during 100 cycles, even under harsh conditions, including an electrolyte amount to cathode capacity (E/C) ratio of **5.58 g Ah^{-1}** , a capacity ratio of the negative electrode to the positive electrode (N/P) of **3.57**, and a low current of **0.1 A g^{-1}** .

Discussion section (Page 14, Line 30 of the revised manuscript):

Consequently, LAPO@Zn|| MnO_2 pouch cells exhibited superior cycle performance, sustaining a high capacity of **3.17 Ah** over 100 cycles under challenging harsh conditions.

Electrochemical measurements section (Page 16, Line 14 of the revised manuscript):

The mass loading of MnO₂ used in coin cell and pouch cell was 1 mg cm⁻² and 4.47 mg cm⁻² (8.94 mg cm⁻² for one single cathode slice as it was double coated), respectively.

Fig. 5 | Electrochemical performance of cells with different electrodes. c, Galvanostatic cycling performance of the LAPO@Zn||MnO₂ pouch cell at 0.1 A g⁻¹. The inset is the optical image of the assembled pouch cell.

Page 39 of the revised Supplementary Information:

Supplementary Note 9. Post-mortem characterizations of the Zn anodes and MnO₂ cathodes extracted from pouch cells.

Supplementary Figure 35. Post-mortem characterizations of the bare Zn anode after 100 cycles in a Zn||MnO₂ pouch cell. a,b, SEM images. **c,** SEM image with

corresponding EDS mapping. **d**, EDX spectrum.

Supplementary Figure 36. Post-mortem characterizations of the LAPO@Zn anode after 100 cycles in a Zn||MnO₂ pouch cell. **a,b**, SEM images. **c**, SEM image with corresponding EDS mapping. **d**, EDX spectrum.

Supplementary Figure 37. Post-mortem characterizations of the MnO₂ cathodes after 100 cycles in bare Zn||MnO₂ and bare LAPO@Zn||MnO₂ pouch cell. **a**, XRD patterns of pristine MnO₂, cycled with bare and LAPO@Zn anodes. **b-g**, SEM images of MnO₂ cycled with LAPO@Zn anodes (b,c), bare Zn anodes (d,e) and pristine MnO₂.

Post-mortem analysis of the anodes and cathodes extracted from pouch cells after 100 cycles provides insight into the degradation mechanisms. The bare Zn anode exhibited

severe damage, including randomly stacked dendrites and significant corrosion (Supplementary Fig. 35a-c), accompanied by elevated oxygen (34.2%) and sulfur (6.1%) levels, suggesting the formation of $Zn_4SO_4(OH)_6 \cdot nH_2O$ byproducts. A detectable silicon signal (2.3%) was also observed (Supplementary Fig. 35d), indicating mechanical damage to the glass fiber separator.

In contrast, the LAPO@Zn anode retained vertically aligned Zn deposits (Supplementary Fig. 36a), with clear remnants of the LAPONITE coating (Supplementary Fig. 36b), and showed significantly lower O (28.9%) and S (3.1%) content (Supplementary Fig. 36d), indicating suppressed corrosion. The presence of Mg (1.6%) and its spatial localization (Supplementary Fig. 36c) confirm the retention of the LAPONITE layer.

On the cathode side, the MnO_2 from the LAPO@Zn-based pouch cell exhibited milder degradation, as indicated by fine flake-like $Zn_4SO_4(OH)_6 \cdot 4H_2O$ deposits (Supplementary Fig. 37b,c) and corresponding XRD signals (Supplementary Fig. 37a). In contrast, the cathode paired with bare Zn showed denser, sheet-like deposits (Supplementary Fig. 37d,e), consistent with the coexistence of $Zn_4SO_4(OH)_6 \cdot 4H_2O$ and $Zn_xMnO(OH)_2$ —an indicator of Mn dissolution. The more complex XRD patterns in the bare Zn group and the dual-phase byproducts suggest stronger pH fluctuations triggered by anode-induced side reactions.

These results indicate that LAPO@Zn effectively suppresses anode-driven corrosion and side reactions, which in turn reduces cathode degradation. Comparing the MnO_2 cathode and LAPO@Zn anode extracted from the same pouch cell, the cathode exhibited more pronounced degradation, suggesting that performance fading in the LAPO@Zn|| MnO_2 pouch cell is predominantly cathode-limited under current conditions.

Figures used in Response Letter only:

Channel information					
CH attribute	Value	CH attribute	Value	CH attribute	Value
Device-unit-cha...	#42-3-1	Voltage range	5V	Aux CH volt. range	—
Start time	2024-04-15 15:32:14	Current range	+30/-30A(6A/30A)	Active material	17442.321mg
Start step ID	1	P/N	2024-04-15 15-32-14	Nominal capacity	

Figure R3. Screenshot of original channel information of the cycling performance of LAPO@Zn||MnO₂ pouch cell at 0.1 A g⁻¹, showing the active material of 17442.321 mg.

Figure R4. Screenshot of original data of cycling performance of LAPO@Zn||MnO₂ pouch cell at 0.1 A g⁻¹, showing the whole discharge capacity evolution during cycling.

Comment 3-5:

While the X-CT imaging and CFD simulations provide strong evidence of ion flux vortex suppression, additional validation via operando electrochemical impedance spectroscopy (EIS) during Zn deposition would further confirm this mechanism.

Response to Comment 3-5:

We sincerely thank Reviewer #3 for this insightful suggestion. In response, we conducted operando electrochemical impedance spectroscopy (EIS) measurements during the initial Zn deposition process in symmetric cells at 1 mA cm⁻². The results are shown in **Supplementary Figs. 29 and 30**.

For LAPO@Zn (**Supplementary Fig. 29**), the initial charge transfer resistance (R_{ct}) rapidly drops from a high value to 246 Ω upon the onset of Zn plating, and stabilizes below 110 Ω from the sixth stage onward (109, 108, 107, 106, and 102 Ω),

indicating fast formation of a uniform and stable electrode-electrolyte interface.

In contrast, bare Zn exhibits a much higher initial R_{ct} of 2250 Ω (**Supplementary Fig. 30**), which decreases slowly and only reaches 231 Ω by the seventh stage. Even in the final stages, the impedance continues to decrease (182, 175, and 164 Ω), suggesting delayed and unstable interfacial evolution.

These results further support our proposed mechanism: the LAPONITE layer facilitates fast suppression of Zn^{2+} - SO_4^{2-} -induced ion flux vortices, leading to stabilized plating behavior. This finding is consistent with our X-ray CT, CFD, and CA analyses.

We are grateful to Reviewer #3 for this valuable suggestion, which has allowed us to gain deeper mechanistic insights into the role of LAPONITE in regulating interfacial ion transport and stabilizing Zn deposition. The following figures and discussion **marked in red** have been added in the **revised Supplementary Information**.

Page 34 of the revised Supplementary Information:

Supplementary Note 6. Operando EIS investigations.

Supplementary Figure 29. Operando EIS analysis of LAPO@Zn symmetric cells at 1 mA cm⁻². a, Galvanostatic discharge curve with 6-min discharge intervals. b, Nyquist plots collected at the start of each segment. c, Evolution of charge transfer resistance (R_{ct}), showing rapid drop and stabilization from the fourth stage onward.

Supplementary Figure 30. Operando EIS analysis of bare Zn symmetric cells at 1 mA cm^{-2} . **a**, Galvanostatic discharge curve with 6-min discharge intervals. **b**, Nyquist plots collected at the start of each segment. **c**, Evolution of charge transfer resistance (R_{ct}), showing rapid drop and stabilization from the fourth stage onward.

The operando EIS results reveal distinct interfacial dynamics between LAPO@Zn and bare Zn. LAPO@Zn exhibits a rapid drop in R_{ct} to 246 Ω , followed by stabilization below 110 Ω after the fourth segment, indicating fast interface stabilization. In contrast, bare Zn starts with 2250 Ω and shows a much slower decline, only reaching 231 Ω by the seventh stage, with further decline afterward.

These results suggest that LAPONITE rapidly suppresses interfacial ion flux vortices and facilitates homogeneous Zn^{2+} transport, leading to steady deposition. The bare Zn interface, lacking this regulation, experiences persistent impedance and delayed stabilization. This operando validation strongly supports our proposed vortex suppression mechanism.

Comment 3-6:

While the X-CT imaging and CFD simulations provide strong evidence of ion flux vortex suppression, additional validation via operando electrochemical impedance spectroscopy (EIS) during Zn deposition would further confirm this mechanism.

Response to Comment 3-6:

We thank Reviewer #3 again for highlighting this important point. This comment seems to be the same as Comment 3-5, and we appreciate your emphasis on the recommendation for operando EIS as additional validation. As suggested, we performed operando EIS measurements, which indeed provide further compelling evidence for ion flux vortex suppression in LAPO@Zn compared to bare Zn. The results are discussed in our response to Comment 3-5 and presented in **Supplementary Note 6** of the **revised Supplementary Information**.

Comment 3-7:

The DFT calculations supporting LAPONITE's charge-separation effects are insightful, but no clear experimental validation of charge redistribution is provided. Techniques such as X-ray photoelectron spectroscopy (XPS) core-level shift analysis or zeta potential measurements could strengthen this claim.

Response to Comment 3-7:

We thank Reviewer #3 for the insightful suggestion. We conducted zeta potential measurements on the LAPONITE+CMC interlayer before and after Zn deposition (1 mA cm⁻² for 1 h), with the latter one peeled from the Zn anode prior to testing. Three replicates were performed for each condition to ensure data reliability. As shown in **Supplementary Table 3**, the zeta potential increased significantly from 1.9764-3.6298 mV (pristine) to 11.9061-13.3090 mV (after deposition).

This increase reflects asymmetric ion redistribution at the LAPONITE surface. Specifically, SO₄²⁻ ions are preferentially anchored within the interlayer spacing of LAPONITE, while Zn²⁺ ions are electrostatically enriched at surface sites. This selective interaction shifts the shear-plane potential to more positive values. The experimental results support our proposed mechanism, in which LAPONITE modulates the local ionic environment by spatially separating Zn²⁺ and SO₄²⁻, suppressing Zn²⁺-SO₄²⁻ contact ion pair formation, and facilitates uniform Zn²⁺ transport. The following

revised contents **marked in red** have been added to the **Material Characterizations section in the revised manuscript** and in the **revised Supplementary Information**.

Page 15, Line 32 of the revised manuscript:

“Zeta potential measurements were carried out using a Malvern Zetasizer Nano ZS.”

Page 30 of the revised Supplementary Information:

Supplementary Note 2. Zeta potential analysis of cation-anion separation behavior in LAPONITE.

The observed increase in zeta potential after Zn deposition, from an average of 2.61 mV in the pristine state to 12.61 mV after deposition (Supplementary Table 3), indicates charge redistribution at the interface. In this measurement, the “after deposition” sample refers to the LAPONITE+CMC interlayer that was peeled from the Zn electrode after 1 h of deposition at 1 mA cm⁻². This change is attributed to selective ion interactions, where Zn²⁺ ions accumulate near the outer surface and SO₄²⁻ ions remain confined within the LAPONITE interlayer. These results support the proposed role of LAPONITE in modulating the local ionic environment by facilitating Zn²⁺ transport and anchoring SO₄²⁻, thereby suppressing Zn²⁺-SO₄²⁻ contact ion pair formation and promoting uniform Zn²⁺ transport.

Supplementary Table 3. Zeta potential of the LAPONITE+CMC interlayer in pristine and post-deposition states (Zn deposition at 1 mA cm⁻² for 1 h).

Sample status	Measurement No.	Zeta Potential (mV)
Pristine	1	1.9764
	2	2.2340
	3	3.6298
After deposition	1	11.9061
	2	12.6010

Comment 3-8:

A. The pouch cell cycling performance is promising, but the long-term degradation mechanisms remain unclear. Post-mortem analysis of cycled anodes (e.g., SEM/EDS mapping after 100 cycles) would be beneficial. B. The pouch cell cycling performance is promising, but the long-term degradation mechanisms remain unclear. Post-mortem analysis of cycled anodes (e.g., SEM/EDS mapping after 100 cycles) would be beneficial. Suggestion: Provide post-cycling characterization of the Zn anode and discuss the advantages of LAPONITE compared to other anode modification strategies.

Response to Comment 3-8:

We sincerely thank Reviewer #3 for this insightful comment. To investigate possible degradation mechanisms, we conducted post-mortem characterizations of the Zn anodes from the Zn||MnO₂ pouch cells after 100 cycles, as shown in **Supplementary Figs. 35 and 36**.

For bare Zn, extensive dendrite growth and randomly stacked structures were observed (**Supplementary Figs. 35a-c**), accompanied by a high oxygen content (34.2%) and noticeable sulfur presence (6.1%) in the EDX spectrum (**Supplementary Figs. 35d**), indicating severe corrosion and formation of byproducts like Zn₄SO₄(OH)₆·nH₂O. Furthermore, the detection of 2.3% Si suggests mechanical damage to the glass fiber separator, likely caused by dendrite penetration.

In contrast, the LAPO@Zn anode exhibits more vertically and orderly aligned Zn clusters (**Supplementary Figs. 36a**) along with discernible remnants of the LAPONITE interlayer (**Supplementary Figs. 36b**). EDS and EDX analyses (**Supplementary Figs. 36c,d**) reveal a lower oxygen content (28.9%) and sulfur content (3.1%) compared to bare Zn, indicating reduced corrosion. The presence of 1.6% Mg further confirms the retention of the LAPONITE layer, highlighting its protective role

during long-term cycling.

Besides, we also investigated the evolution of MnO₂ cathodes extracted from pouch cells. We have included the figures and related discussion **marked in red** below in the **revised Supplementary Information** for better clarity.

Page 39 of the revised Supplementary Information:

Supplementary Note 9. Post-mortem characterizations of the Zn anodes and MnO₂ cathodes extracted from pouch cells.

Supplementary Figure 35. Post-mortem characterizations of the bare Zn anode after 100 cycles in a Zn||MnO₂ pouch cell. **a,b**, SEM images. **c**, SEM image with corresponding EDS mapping. **d**, EDX spectrum.

Supplementary Figure 36. Post-mortem characterizations of the LAPO@Zn anode after 100 cycles in a Zn||MnO₂ pouch cell. **a,b**, SEM images. **c**, SEM image with corresponding EDS mapping. **d**, EDX spectrum.

Supplementary Figure 37. Post-mortem characterizations of the MnO₂ cathodes after 100 cycles in bare Zn||MnO₂ and bare LAPO@Zn||MnO₂ pouch cell. **a**, XRD patterns of pristine MnO₂, cycled with bare and LAPO@Zn anodes. **b-g**, SEM images of MnO₂ cycled with LAPO@Zn anodes (b,c), bare Zn anodes (d,e) and pristine MnO₂.

Post-mortem analysis of the anodes and cathodes extracted from pouch cells after 100 cycles provides insight into the degradation mechanisms. The bare Zn anode exhibited severe damage, including randomly stacked dendrites and significant corrosion (Supplementary Fig. 35a-c), accompanied by elevated oxygen (34.2%) and sulfur (6.1%) levels, suggesting the formation of Zn₄SO₄(OH)₆·nH₂O byproducts. A detectable silicon signal (2.3%) was also observed (Supplementary Fig. 35d), indicating mechanical damage to the glass fiber separator.

In contrast, the LAPO@Zn anode retained vertically aligned Zn deposits (Supplementary Fig. 36a), with clear remnants of the LAPONITE coating (Supplementary Fig. 36b), and showed significantly lower O (28.9%) and S (3.1%) content (Supplementary Fig. 36d), indicating suppressed corrosion. The presence of Mg (1.6%) and its spatial localization (Supplementary Fig. 36c) confirm the retention of the LAPONITE layer.

On the cathode side, the MnO₂ from the LAPO@Zn-based pouch cell exhibited

milder degradation, as indicated by fine flake-like $\text{Zn}_4\text{SO}_4(\text{OH})_6 \cdot 4\text{H}_2\text{O}$ deposits (Supplementary Fig. 37b,c) and corresponding XRD signals (Supplementary Fig. 37a). In contrast, the cathode paired with bare Zn showed denser, sheet-like deposits (Supplementary Fig. 37d,e), consistent with the coexistence of $\text{Zn}_4\text{SO}_4(\text{OH})_6 \cdot 4\text{H}_2\text{O}$ and $\text{Zn}_x\text{MnO}(\text{OH})_2$ —an indicator of Mn dissolution. The more complex XRD patterns in the bare Zn group and the dual-phase byproducts suggest stronger pH fluctuations triggered by anode-induced side reactions.

These results indicate that LAPO@Zn effectively suppresses anode-driven corrosion and side reactions, which in turn reduces cathode degradation. Comparing the MnO_2 cathode and LAPO@Zn anode extracted from the same pouch cell, the cathode exhibited more pronounced degradation, suggesting that performance fading in the LAPO@Zn|| MnO_2 pouch cell is predominantly cathode-limited under current conditions.

In terms of comparison with other Zn anode modification strategies, our LAPONITE coating offers several distinct advantages. Existing strategies generally fall into three categories:

1. Versus general artificial coatings (e.g., Nano Res. 2024, 17, 8104):

Unlike common coatings that primarily serve as physical barriers, LAPONITE offers unique cation-anion separation capability, effectively alleviating ion flux vortices, as supported by our CFD simulation (**Fig. 2e**) in contrast with general coatings (**Supplementary Fig. 7d**).

2. Versus bulk electrolyte modifications, including sol electrolytes (Nat. Commun. 2023, 14, 4211; Angew. Chem. Int. Ed. 2024, 63, e202401441) and molecular crowding (ACS Nano 2023, 17, 23207):

While these strategies contribute to suppressing water decomposition reactions, they may also introduce side effects on cathode performance and complicate electrolyte formulation. In contrast, our system retains a dilute and unmodified aqueous electrolyte, which avoids such interference while enabling fast interfacial reaction kinetics due to reduced Zn^{2+} desolvation barriers in aqueous media (Energy Environ. Sci. 2018, 11,

881).

3. Versus electrolyte additives (e.g., Nat. Nanotechnol. 2021, 16, 902; Angew. Chem. Int. Ed. 2022, 61, e202212839):

These additives cannot effectively occupy the inner Helmholtz layer and thus have limited ability to inhibit side reactions or guide Zn deposition, unlike the complete and stable coverage provided by the LAPONITE layer in our work.

Moreover, the latter two strategies not only provide insufficient protection for the Zn anode, but may also negatively impact cathode performance. As a result, they are often overlooked in full-cell design and tend to lead to less stable cycling in practical configurations compared to LAPO@Zn.

We thank the reviewer again for encouraging us to further clarify the distinct benefits of our approach, which has helped us gain deeper insight into the unique role of the LAPONITE coating layer.

Comment 3-9:

Figure 1: The color scheme for Zn and SO_4^{2-} separation should be more distinct to enhance readability. The schematic in Fig. 1b should include charge labels to clarify Zn^{2+} and SO_4^{2-} behavior.

Response to Comment 3-9:

We sincerely thank Reviewer #3 for this valuable suggestion. In the original version, the color of SO_4^{2-} was too similar to that of the LAPONITE lamella, which compromised visual clarity. In the revised figure, we have changed the SO_4^{2-} color to periwinkle to make it more distinguishable. We also added charge labels (“+” and “-”) to the Zn^{2+} and SO_4^{2-} spheres, respectively, and marked “CIP” inside the contact ion pair icons to explicitly indicate their identities. These revisions aim to improve clarity and better illustrate the charge separation behavior. We hope the updated figure now meets the high presentation standards of *Nature Communications*. The new Fig. 1b marked in red in the updated manuscript is shown as follow:

LAPONITE separates Zn^{2+} and SO_4^{2-}

Fig. 1 | Cation/anion separation effect of LAPONITE. b, Schematic illustration of the penetration of ions through the LAPONITE interlayer on substrates. The grey, **periwinkle** and **muted navy** balls represent Zn^{2+} , SO_4^{2-} , and $Zn^{2+}-SO_4^{2-}$ CIP, respectively.

Comment 3-10:

Figure 2: CFD streamline plots (Fig. 2a,c) lack a scale bar or color legend to indicate flow intensity. Clarify what simulation parameters were used in CFD to avoid ambiguity.

Response to Comment 3-10:

We thank Reviewer #3 for the valuable feedback. We apologize for the confusion caused by the streamline plots in **Fig. 2a and 2c**. These images were not generated using COMSOL's streamline module and do not represent actual velocity fields. Instead, they are schematic illustrations based on the simulated steady-state Zn^{2+} concentration gradients, intended to qualitatively compare ion flux behavior near uncoated and LAPONITE-coated Zn surfaces.

Since the plots serve only as conceptual representations rather than quantitative outputs, no scale bar or color legend was included. To prevent misunderstanding, we have revised the caption of **Fig. 2** to clarify that the streamlines are illustrative and not physically computed. For quantitative analysis, **Fig. 2d and 2e** report residual sum of squares (RSS) values and stimulated concentration gradients. Furthermore, the absence of visible vortices in **Fig. 2c** is attributed to reduced $Zn^{2+}-SO_4^{2-}$ contact ion pairing,

which suppresses chaotic convection and supports more uniform Zn^{2+} flux distribution.

We have also added a detailed description of the CFD simulation setup, equations, parameters, and boundary conditions in the revised Supplementary Information (Supplementary Note 11), and made corresponding adjustments marked in red in the revised manuscript to ensure clarity and reproducibility.

Page 5, Line 9 of the revised manuscript:

The resulting imbalance in mass transfer near the surface induces significant ion flux vortices and disrupts the uniformity of Zn deposition sites, as illustrated conceptually in the schematic streamlines of Fig. 2a.

Page 6 of the revised manuscript:

Fig. 2 | Diffusion progress on different electrodes. a,c, Schematic of streamline upon deposition substrate without (a) and with LAPONITE (c). b, Force analysis of Zn^{2+} , SO_4^{2-} , and $Zn^{2+} - SO_4^{2-}$ CIP under applied electric field.

Page 44 of the revised Supplementary Information:

Supplementary Note 11: CFD simulation method and parameters for Fig. 2.

The following equations and parameters were used in the simulations.

1. Governing Equations:

Navier-Stokes equations for incompressible fluid motion in the electrolyte (Ω_1).

Darcy's law for the porous LAPONITE coating (Ω_2), with Kozeny-Carman model-

derived forces
$$F_{porous} = \frac{180\mu(1-\epsilon)^2}{dp^2\epsilon^3}u + \frac{\rho}{\epsilon^2}\nabla \cdot u$$

Convection-diffusion equations for ion transport, incorporating free (c_{free}) and adsorbed (c_{abso}) ion concentrations.

2. Key Parameters:

- Porosity (ϵ): 1 (electrolyte) and 0.3 (porous LAPONITE layer).
- Nanopore diameter (dp): 1 nm.
- Diffusion coefficients: $D_1=0.8\times 10^{-7} \text{ m}^2 \text{ s}^{-1}$ (electrolyte), anisotropic D_1 in the coating (enhanced along the channel axis), and $D_2\approx 10^{-11} \text{ m}^2 \text{ s}^{-1}$ for adsorbed ions.
- Boundary conditions: Nonzero velocity at the top boundary to induce artificial vortices, zero velocity elsewhere; fixed ion concentration at the top ($c_{\text{free}}=\text{initial value}$), zero flux at side boundaries.

3. **Domain:** 2D region ($100 \times 100 \mu\text{m}^2$) above the electrode.

Comment 3-11:

Figure S22: Zn||MnO₂ or LAPO@ Zn||MnO₂ which one is correct?

Response to Comment 3-11:

We thank Reviewer #3 for pointing out this important clarification. The cell configuration shown in original **Supplementary Fig. 22** is indeed LAPO@Zn||MnO₂, not bare Zn. This experiment was designed to demonstrate the performance improvement enabled by the LAPO@Zn anode under relatively high current density (2 A g^{-1}). To eliminate any confusion, we have now revised the figure caption accordingly. The corrected figure now appears as **Supplementary Fig. 23** in the **revised Supplementary Information** (due to renumbering after insertion of newly added figures), as shown below:

Page 24 of the revised Supplementary Information:

Supplementary Figure 23. Galvanostatic cycling performance of LAPO@Zn||MnO₂ coin cells at 2 A g⁻¹.